# The *Arabidopsis* SGN3/GSO1 receptor kinase integrates soil nitrogen status into shoot development

Defeng Shen [iD][1], Kathrin Wippel[1,2,5], Simone Remmel[1], Yuanyuan Zhang[1], Noah Kuertoes[1], Ulla Neumann[3], Stanislav Kopriva [iD][2,4] & Tonni Grube Andersen [iD][1,2]✉

## Abstract

**The Casparian strip is a barrier in the endodermal cell walls of plants that allows the selective uptake of nutrients and water. In the model plant *Arabidopsis thaliana*, its development and establishment are under the control of a receptor-ligand mechanism termed the Schengen pathway. This pathway facilitates barrier formation and activates downstream compensatory responses in case of dysfunction. However, due to a very tight functional association with the Casparian strip, other potential signaling functions of the Schengen pathway remain obscure. In this work, we created a MYB36-dependent synthetic positive feedback loop that drives Casparian strip formation independently of Schengen-induced signaling. We evaluated this by subjecting plants in which the Schengen pathway has been uncoupled from barrier formation, as well as a number of established barrier-mutant plants, to agar-based and soil conditions that mimic agricultural settings. Under the latter conditions, the Schengen pathway is necessary for the establishment of nitrogen-deficiency responses in shoots. These data highlight Schengen signaling as an essential hub for the adaptive integration of signaling from the rhizosphere to aboveground tissues.**

**Keywords** Casparian Strip; Receptor Signaling; Nitrogen Responses; Nutrient Physiology; Root Development
**Subject Category** Plant Biology

## Introduction

To survive in highly dynamic environments, plants deploy hydrophobic barriers to protect their vital tissues (Geldner, 2013). One particularly well-studied example is the Casparian strip (CS), which in roots is situated in the endodermal cell layer. The CS blocks flow within the extracellular matrix and forces solute uptake to occur selectively across the endodermal plasma membrane (PM) (Barberon and Geldner, 2014). In most plants, including the model plant *Arabidopsis thaliana* (hereafter Arabidopsis), the CS consists of cell-spanning, lignified "bands" in the anticlinal cell walls. These are formed in the extracellular regions adjacent to the so-called Casparian strip domain (CSD) in the PM (Roppolo et al, 2011). The CSD is established in differentiating endodermal cells by the scaffold-like CASPARIAN STRIP MEMBRANE PROTEINs (CASPs) and a number of secreted enzymes responsible for CS lignification (Baxter et al, 2009; Hosmani et al, 2013). Expression of most genes involved in CS formation (e.g., *CASPs*) is controlled by the R3R2 MYB-class transcription factor MYB36 (Kamiya et al, 2015; Liberman et al, 2015). Functional barrier establishment additionally requires a signaling mechanism known as the Schengen (SGN) pathway—a receptor-ligand system that co-occurs independently, but in concert with the onset of CS establishment (Pfister et al, 2014) and faciliates fusion of CS depositions in the apoplast into a coherent barrier (Doblas et al, 2017; Fujita et al, 2020). The main known components of the SGN pathway are the stele-synthesized CASPARIAN STRIP INTEGRITY FACTOR (CIF) peptide ligands, their receptor-kinase target SCHENGEN3/GASSHO1 (SGN3/GSO1) and the downstream kinase SCHENGEN1 (SGN1/PBL15) (Doblas et al, 2017; Fujita, 2021; Pfister et al, 2014). As CIF peptides diffuse between endodermal cells, they induce activation of SGN3, which results in functional barrier formation and thereby completes a self-regulating system that prevents further diffusion of CIF peptides (Doblas et al, 2017; Nakayama et al, 2017). Ectopic CIF treatment, or mutants where CS function is disrupted (e.g., *myb36* knockouts), leads to over-activation of the SGN pathway and initiates a number of responses such as ectopic lignification and increased suberin deposition (Fujita, 2021). This CS fusion and "surveillance" mechanism remain the main known functions of SGN3 despite the resemblance of this receptor to other multi-purpose signaling integrators (Bender and Zipfel, 2023).

CS-related responses initiated by the SGN pathway are remarkably effective in compensating abiotic (Pfister et al, 2014;

[1]Department of Plant-Microbe Interactions, Max Planck Institute for Plant Breeding Research, Carl-von-Linné-Weg 10, 50829 Cologne, Germany. [2]Cluster of Excellence on Plant Sciences (CEPLAS), Cologne, Germany. [3]Central Microscopy, Max Planck Institute for Plant Breeding Research, Carl-von-Linné-Weg 10, 50829 Cologne, Germany. [4]Institute for Plant Sciences, University of Cologne, Zülpicher Str. 47b, 50674 Cologne, Germany. [5]Present address: Swammerdam Institute for Life Sciences, University of Amsterdam, Science Park 904, 1098XH Amsterdam, The Netherlands. ✉E-mail: tandersen@mpipz.mpg.de

Reyt et al, 2021) as well as biotic (Salas-González et al, 2021) consequences of a dysfunctional barrier. Common to most mutants with disturbed function of CS are changes in the ionic profile of shoots— in particular with respect to potassium (K), which is assumed to leak out from the roots due to lack of stelar retention. However, recent evidence indicates that the SGN pathway provides signaling for a sensing mechanism of local potassium (K) status in the young root via a ROS/Ca$^{2+}$-dependent signaling mechanism (Wang et al, 2021). This depends on the NADPH oxidases RESPIRATORY BURST OXIDASE HOMOLOG D and F (RBOHD and F) (Fujita et al, 2020). Alongside with this, mutants with an activated SGN pathway show induced abscisic acid (ABA)-related responses in the shoots (Wang et al, 2019), which combined implicates the SGN system in systemic integration of root responses. Yet, lack of mutants that have uncoupled SGN signaling and CS function makes it difficult to assess whether such physiological responses directly depend on the barrier function or are a signaling-coupled consequence of changed SGN activation. Moreover, as most of our current understanding of the CS-SGN system comes from plants grown in axenic agar plate conditions, it remains to be evaluated how this system integrates root responses into the shoots under more agriculturally relevant conditions.

In this work, we created a mechanism to uncouple the CS status from SGN signaling by a semi-synthetic rewiring of the genetic network that underlies CS formation. This resulted in a new class of endodermal barrier mutants with earlier SGN-independent CS formation, which we were able to hold against characterized mutants. Through an integrative experimental setup, we performed a direct evaluation of increased CS formation and SGN signaling individually. Our work provides evidence that signaling rather than barrier formation serves to integrate soil status between above- and below-ground tissues. Combined, this brings forth a model where the SGN pathway serves a pivotal role in integrating CS establishment, rhizosphere status and nutrient-related transcriptional reprogramming across the entire plant.

## Results

### Rewiring MYB36 creates an earlier SGN-independent Casparian strip

To create plants which uncouple the function of SGN pathway from CS fusion, we employed the promoter region of the direct MYB36 target *CASP1* (Kamiya et al, 2015) to drive *MYB36* expression. Our reasoning was that this generates a spatially restricted expression feedback where MYB36 induces itself and its downstream targets only in cells that have initiated *CASP1* expression and thus are already differentiated toward CS formation (MYB36$_{Loop}$). This avoids pleiotropic effects associated with barrier establishment outside the endodermis, by also overrides additional inputs needed for CS fusion such as SGN3 activation. To evaluate functionality of this idea, we performed a whole-root transcriptome analysis of two independent homozygous lines carrying the MYB36$_{Loop}$ construct and compared them to the *myb36-2* mutant normalized to their respective parental lines (Col-0 for *myb36-2* and pCASP1::CASP1-GFP for MYB36$_{Loop}$ lines). Presence of MYB36$_{Loop}$ led to strongly increased expression of both *MYB36* and *CASP1*, which were almost non-detectable in the *myb36-2*

mutant (Fig. 1A). Within our significance threshold (false recovery rate (FDR) < 0.05, |Log$_2$ FC| > 2), a subset of 113 differentially expressed genes (DEGs) showed a similar response as *CASP1* and *MYB36*, whereas 142 showed the opposite behavior (i.e repressed in MYB36$_{Loop}$ and induced in *myb36-2*) (Fig. EV1A). Within the first set, we found most genes with a characterized function in CS establishment (including components of the SGN pathway) (Fig. 1B), which indicates that the MYB36$_{Loop}$ plants have an increased expression of the CS-forming machinery. In line with this, the functional gene ontology (GO) term "cell–cell junction assembly" was overrepresented among the genes induced in MYB36$_{Loop}$ and repressed in *myb36-2*, respectively (Fig. EV1B). Plants expressing MYB36$_{Loop}$ had an almost tripled anticlinal CS width when compared to WT (from ~500 nm in WT to an average of approximately 1500 nm in MYB36$_{Loop}$ plants), which illustrates that these transcriptional changes were directly reflected in the formation of CS (Fig. 1C,D). Interestingly, this increase in CS width was accompanied by an earlier onset of a functional CS formation (Fig. 1E). This can be interpreted as either an increased CS deposition rate or as a faster maturation of CS as we did not observe any changes in the initiation of CS depositions or xylem development (Fig. EV1D). Increased CS deposition would not affect the spatial activation of the SGN system, but early-onset functional CS would in theory limit the zone in which diffusion of CIF peptides can occur. In line with this, GO terms associated with an activated SGN system were repressed in MYB36$_{Loop}$ (Fig. EV1A,B). Moreover, only one CS-related gene, *CASP4*, requires both MYB36 and a functional SGN system for expression (Fig. EV1C) and particularly this gene was repressed in the MYB36$_{Loop}$ lines (Fig. 1B). Moreover, when introducing MYB36$_{Loop}$ into *sgn3-3* mutants, the defective CS found in this line (Pfister et al, 2014) was fully complemented by the self-reinforcing *MYB36* expression (Fig. 1F,G). With basis on these findings, we conclude that the loop-driven ectopic expression of *MYB36* indeed uncouples the barrier formation from its endogenous SGN-dependency and creates an earlier onset of a functional CS barrier. MYB36$_{Loop}$ plants did not show ectopic CS formation in adjacent cell types (Fig. EV1E), which supports that this feedback driven expression was contained to the endodermis.

### Overloading the Casparian strip domain leads to ectopic CS-like structures

The introduction of a positive feedback loop into the CS formation machinery should in principle lead to an exponential increase in CS-producing components. Therefore, to investigate the dynamics of CS establishment, the MYB36$_{Loop}$ lines were generated in a pCASP1::CASP1-GFP background (Roppolo et al, 2011). The onset of CASP1-GFP expression has been characterized to follow a "string-of-pearls" pattern in the anticlinal PM of differentiating endodermal cells (Roppolo et al, 2011). This was also the case for the MYB36$_{Loop}$ containing plants (Fig. 2A). However, ~2 h after onset, the CASP1-GFP signal split into two laterally expanding lines that likely represent the increased CS width (Fig. 1C,D). Intriguingly, after about 7 h, the CASP1-GFP signal formed "ring-like" radially expanding patches in the periclinal cell walls, which normally do not host CS formation (Fig. 2A,B; Movie EV1). These structures contained lignin-specific signals (Fig. 2B), which supports that, besides CASP1-GFP, the entire lignification program

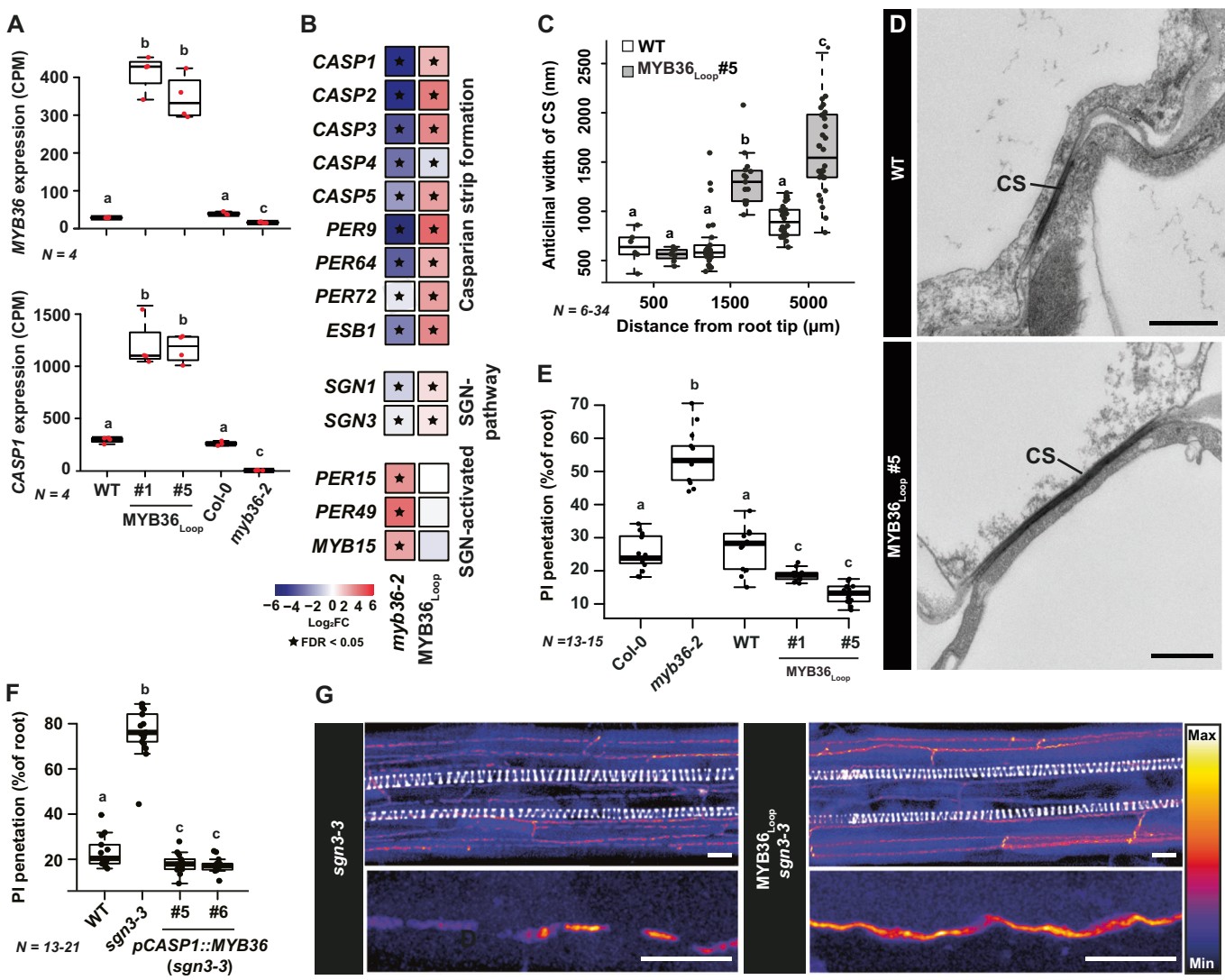

**Figure 1. Transcriptional responses and Casparian strip formation in plants expressing MYB36 driven by the *CASP1* promoter.**

(A) Normalized read counts of *MYB36* and *CASP1* (CPM). (B) Heatmap depicting expression of individual genes from transcriptome analysis of whole roots. Star symbols represent significantly differentially expressed (FDR < 0.05) genes compared with the corresponding parental lines. (C) Quantification of Casparian strip width in transmission electron microscopy (TEM) images. (D) TEM images of Casparian strips (pointed by black lines) between two endodermal cells, scale bar: 500 nm. (E) Quantification of onset of propidium iodide blockage. Combined data from two independent experiments. (F) Measurement of the percentage of roots that can be penetrated by propidium iodide (PI). Combined data from two independent experiments. (G) Maximum projection of a confocal image stack of Basic Fuchsin-stained 7-day-old roots. Scale bars in upper images represent 10 µm, whereas scale bars in lower images represent 5 µm. For boxplots, the center line indicates median, dots represent data points, the box limits represent the upper and lower quartiles, and whiskers maximum and minimum values. WT represents the parental line (pCASP1:CASP1-GFP) of MYB36$_{Loop}$ plants. Numbers of biological replicates are indicated on graph. Letters depict statistical differences in a one-way ANOVA analysis with a Holm–Sidak-adjusted post hoc *t* test (*P* < 0.05). CS Casparian strip, CPM counts per million, PI propidium iodide. Source data are available online for this figure.

responsible for CS polymerization was present. To test this, we created combinatorial lines that expressed the MYB36$_{Loop}$ construct, the fluorescent marker for CASP1-GFP and the lignifying CS enzymes ESB1-mCherry or PER64-mCherry. Indeed, for both enzymes, the markers co-occurred with the ectopic CASP1-GFP patches, although in a broader zone that extended beyond the CASP1-GFP signal (Figs. 2C and EV1F). In combination with the expanding nature of the CASP1-GFP signal (Movie EV1), this suggests that ESB1-dependent lignin polymerization (Gao et al, 2023) may occur at the edge of the CS and expand outwards. Interestingly, MYB36$_{Loop}$ plants had a slight

delay of endodermal suberization (Fig. EV1H), and suberin was excluded from the lignified patches within the individual cells (Fig. EV1G). Treatment with 100 nM CIF2 had no effect on the suberin patterning in MYB36$_{Loop}$ lines, but induced "fringe-like" depositions in the ectopic periclinal CS depositions (Fig. EV1H–J). Combined, these findings confirm an intriguing recently found connection between the CS enzyme ESB1 and lignification of the CS. The dynamic changes in the subcellular organization of CS formation can form the basis for a deeper understanding of the mechanisms underlying the specific localization of CS to the anticlinal cell wall.

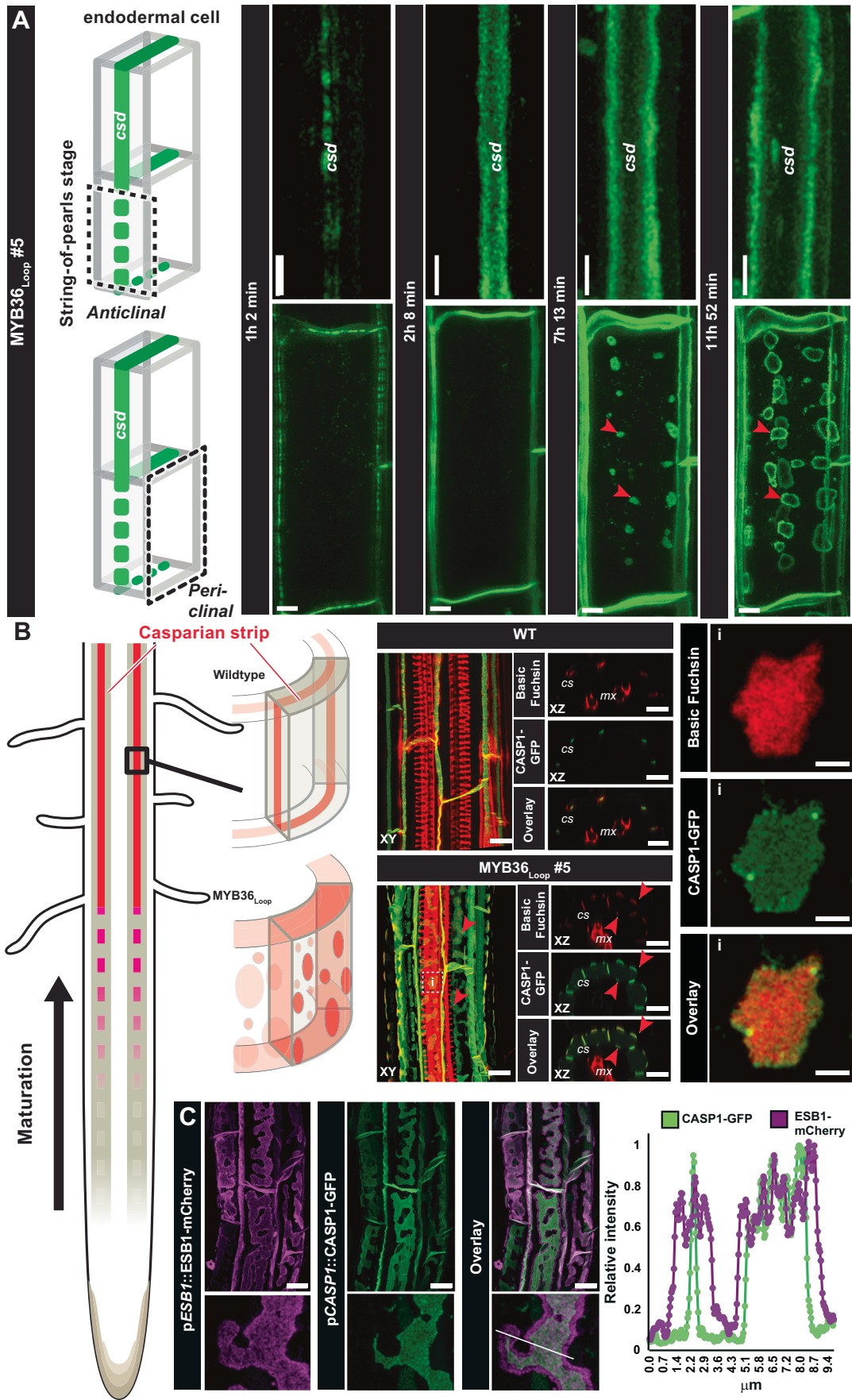

**Figure 2.   Analysis of Casparian strip onset and Schengen responses in plants expressing MYB36 driven by the *CASP1* promoter.**

(A) Time course of CASP1-GFP expression in MYB36$_{Loop}$#5 plants. Imaging was initiated at the string-of-pearls stage before onset of CASP1-GFP expression. Images are a maximum projection of confocal image stacks from the anticlinal (upper graph) and periclinal (lower graph) view of an individual endodermal cell. Casparian strip domain (CSD), red arrowheads point to ectopic CASP1-GFP in the periclinal plasma membrane. (B) Maximum projections of confocal image stacks of Basic fuchsin-stained wild-type and MYB36$_{Loop}$#5 roots expressing the CASP1-GFP fusion reporter in the region where mature CS is formed. Arrowheads represent ectopic lignin-specific signals on the periclinal cell walls of endodermis in MYB36$_{Loop}$#5 roots. For the insert (i) scale bar: 2 μm. (C) Maximum projection of a confocal image stack of Basic fuchsin-stained 7-day-old MYB36$_{Loop}$ roots expressing pESB1::ESB1-mCherry and pCASP1::CASP1-GFP. Line in overlay depicts the transect used for relative intensity measurements. Unless otherwise stated, the scale bars represent 10 μm (A–C). Source data are available online for this figure.

## Increased Casparian strip formation provides abiotic stress resistance

Intrigued by the changed CS formation observed in the MYB36$_{Loop}$ plants, we set out to evaluate how this affects responses to abiotic stress. Under standard agar-based conditions, only MYB36$_{Loop}$ plants showed a significantly reduced primary root length when compared to their parental line (Figs. 3A and EV2A). In both of our selected lines as well as the *myb36-2* mutant, this was accompanied by a significant reduction in lateral root (LR) density (Figs. 3B and EV2A). In *myb36-2* LR repression is related to changes in ROS formation (Fernández-Marcos et al, 2017), yet the MYB36$_{Loop}$ lines had an earlier repression of primordia development than *myb36-2* (Fig. EV2B) and these appeared "flattened" against the endodermis (Fig. EV2C). This is therefore most likely due to the increased CS deposition creating a mechanical hindrance that physically inhibits root emergence. When subjected to salt or osmotic stress, MYB36$_{Loop}$ plants had improved primary root growth compared to their parental line (Fig. 3C). This proposes an increased tolerance to abiotic stress, which is consistent with observations describing that functional CS formation occurs earlier in the young root under certain abiotic stress conditions (Salas-González et al, 2021; Wang et al, 2021). Under nutrient-rich (½ MS) conditions, MYB36$_{Loop}$ expressing roots displayed reduced activity of genes encoding for nitrogen/phosphorus-related stress responses such as *NIN-LIKE-Proteins* (*NLPs*) (Konishi and Yanagisawa, 2013) as well as a concurrent induction of repressors belonging to the *NITRATE-INDUCIBLE, GARP-TYPE TRANSCRIPTIONAL REPRESSORs* (*NIGTs*) (Kiba et al, 2018) and *BTB AND TAZ DOMAIN PROTEINs* (*BTs*) families (Araus et al, 2016) (Fig. 3D). As these genes respond to nitrogen- and phosphorus-related stress, this suggests that the MYB36$_{Loop}$ plants are either affected in signaling associated with these stresses or display increased nitrogen/phosphorus accumulation by preventing backflow to the media. Indeed, when germinated in the absence of nitrogen or phosphorus, MYB36$_{Loop}$ plants displayed an increased relative root growth when compared to their parental line while we found no significant changes were observed in *myb36-2* plants (Fig. EV2D). We found no changes in the size of the shoots, nor any significant changes in media lacking sulfur (Fig. EV2D). Interestingly, the onset of CS barrier formation responds to nitrogen status in maize (*Zea mays*) (Guo et al, 2023) and we therefore analyzed if similar effects can be observed in Arabidopsis. Here, WT plants showed a dose-dependent delay of CS function specifically when nitrogen but not phosphorus supply was restricted, which was not observed in MYB36$_{Loop}$ plants (Fig. 3E,F). Taken together, this supports the well-established idea that apoplastic blockage confers increased resistance to abiotic stresses, but emphasizes the existence of an

intriguing dynamic role of barriers under nitrogen starvation which may be disabled in the MYB36$_{Loop}$ plants due to the early-onset CS formation.

### Manipulation of the CS-SGN system disturbs shoot K-homeostasis in a soil-independent manner

Next, to test how MYB36$_{Loop}$ presence affects growth responses in more complex situations such as soil, we measured shoot performance under different soil conditions. We used standard potting soil to represent a nutrient-rich environment and compared these with plants grown on a local agricultural soil (Cologne Agricultural Soil, CAS). Independent of soil type, *myb36-2* and MYB36$_{Loop}$ rosettes were both significantly smaller than their parental lines (one-way ANOVA, $P < 0.05$) (Fig. 4A,B). As barrier mutants show characteristic changes in their shoot ionome (Salas-González et al, 2021), we also analyzed the mineral ion content of shoots across the two employed soil types. In both soils, MYB36$_{Loop}$ and *myb36-2* rosettes accumulated distinct mineral profiles when compared to each other as well as their parental lines (FDR < 0.05) (Figs. 4C and EV4A; Datasets EV1 and EV2). However, a large part of the variation in mineral content could be explained by the soil type (Fig. 4C). The strongest effects were observed when plants were grown on CAS, where most heavy metals were increased in MYB36$_{Loop}$ plants but decreased in *myb36-2* (Figs. 4D and EV4A) and thus likely a direct effect of the opposing CS status in these mutants. Yet, independent of soil type, both MYB36$_{Loop}$ and *myb36-2* rosettes contained significantly decreased amounts of potassium (K) (one-way ANOVA, $P < 0.01$) (Figs. 4D and EV3A). Low K accumulation in shoots consistently correlates with ineffective barrier formation (Pfister et al, 2014; Salas-González et al, 2021) and it was therefore unexpected to find a reduction in K content in MYB36$_{Loop}$ plants. Thus, despite the soil type being the strongest driver for shoot mineral content, shoot accumulation of K appears to include SGN-dependent regulatory components.

## Shoots display distinct SGN- and barrier-dependent transcriptional responses

To dig deeper into the observed mineral differences, we measured anionic nutrient content in the shoots. In all genotypes, phosphate, and sulfate showed a slight reduction when plants grown on CAS when compared to potting conditions (Fig. EV4B). In contrast to this, we found that all rosettes had strongly reduced levels of nitrate on CAS (Fig. 4E). Indeed, all parental lines and *myb36-2* plants from CAS conditions showed signs of stress-induced anthocyanin accumulation consistent with nitrogen starvation (Chalker-Scott, 1999; Diaz et al, 2006). Remarkably, this was not observed in MYB36$_{Loop}$ rosettes in either WT or *sgn3-3* backgrounds, which

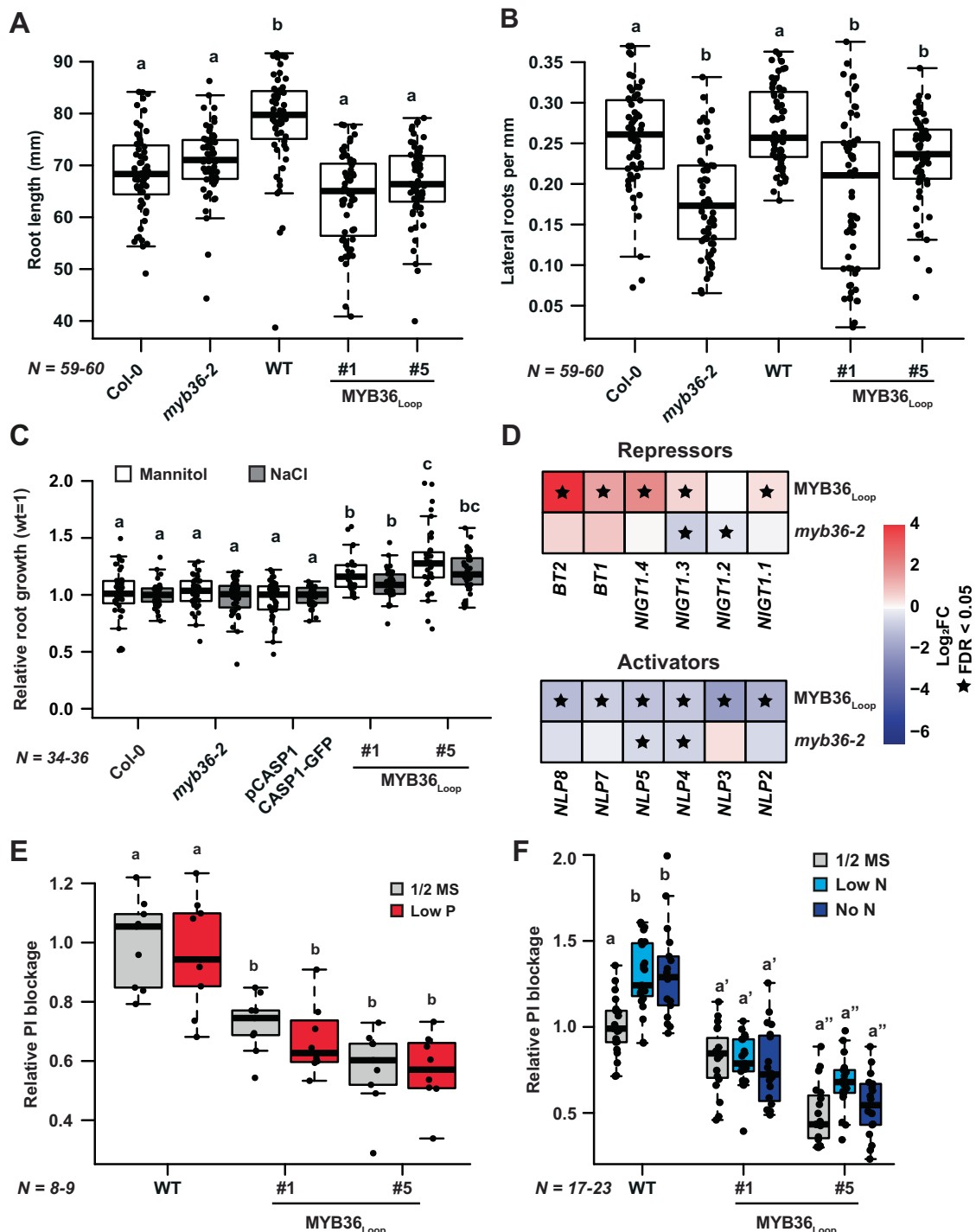

remained green under these conditions (Figs. 4A and EV4). A similar phenotype was observed in the *sgn3-3* mutant, which, besides impaired SGN signaling, has a dysfunctional CS (Pfister et al, 2014) rather than the increased and earlier onset in MYB36Loop plants or the compensatory SGN activation found in *myb36-2* (Figs. 4A and 5A). This proposes a so-far overlooked involvement of the SGN pathway rather than the CS in shoot nitrogen-starvation responses. To further investigate this, we

performed a transcriptional analysis of rosettes from plants grown on different soils. Among the 253 DEGs specifically repressed in MYB36Loop and *sgn3-3* under CAS conditions (Fig. EV4C), functions related to nitrogen starvation, leaf senescence, salt stress and salicylic acid were specifically enriched (Fig. EV4D). Under potting soil conditions, there were only a handful of genes present in the same overlap (Fig. EV4E). Combined, the observed transcriptional responses indicate that the MYB36Loop and *sgn3-3*

Figure 3. Root architecture and abiotic stress analysis in mutants with modified Casparian strip formation.

(A) Primary root length of 14-day-old plants from ½ MS agar. Combined data from three independent experiments. (B) Lateral root density of 14-day-old plants grown on ½ MS agar. Combined data from three independent experiments. (C) Relative root growth of 14-day-old plants after 7 days under stress conditions (150 mM mannitol or 100 mM NaCl). Combined data from two independent experiments. (D) Heatmap of genes belonging to the BTB AND TAZ DOMAIN PROTEINS (BT), NITRATE-INDUCIBLE, GARP-TYPE TRANSCRIPTIONAL REPRESSORs (NIGT) or NIN-LIKE Protein (NLP) families in roots of MYB36$_{Loop}$ and myb36-2 plants. Star symbols represent significantly differential expressed (FDR < 0.05) genes. (E) Onset of functional Casparian strip by means of propidium iodide (PI) blockage. ½ MS agar medium (½MS) or 100 µM P (Low P). (F) Onset of functional Casparian strip by means of propidium iodide (PI) blockage. ½ MS agar medium (½MS), 0.11 mM N (Low N) or 0 mM N (No N). Combined data from two independent experiments. For boxplots, the center line in the box indicates median, dots represent data, limits represent upper and lower quartiles, and whiskers maximum and minimum values. Different letters depict statistical difference in a one-way ANOVA analysis with a Holm–Sidak-adjusted post hoc $t$ test ($P < 0.05$). WT represents the parental line (pCASP1:CASP1-GFP) of MYB36$_{Loop}$ plants. #1 and #5 refer to two independent homozygous lines of MYB36$_{Loop}$ plants. Numbers of biological replicates are indicated on graph. Source data are available online for this figure.

mutants are repressed in their ability to respond correctly to soil nitrogen conditions. Both MYB36 and SGN3 expression was barely above the signal detection threshold in rosettes (Fig. EV4F). In conclusion, responses coordinated across tissues by the SGN pathway are soil-status dependent and independent of CS status in the root endodermis.

### Shoot nitrogen-starvation responses are dependent on CIF activation of SGN3 in roots

Among the responses specific to MYB36$_{Loop}$, sgn3-3 or the combination of the two (thus related to SGN3 signaling rather than CS status), we found induction of the nitrogen-starvation signal repressor genes belonging to the *NITRATE-INDUCIBLE, GARP-TYPE* (*NIGT*) family (Araus et al, 2016; Kiba et al, 2018) as well as *LATERAL BOUNDARY DOMAIN 37, 38, and 39* (*LBD37, 38, and 39*) (Rubin et al, 2009) (Fig. 4F). This was intriguing since, under nitrogen-sufficient conditions, LBD37-39 repress expression of the anthocyanin master regulators *PRODUCTION OF ANTHO-CYANIN PIGMENT 1* and *2* (*PAP1* and *PAP2*) thereby prevent anthocyanin accumulation (Li et al, 2018; Rowan et al, 2009; Rubin et al, 2009). Indeed, *PAP1* and *PAP2* as well as their targets (Dooner et al, 1991) were repressed in MYB36$_{Loop}$, sgn3-3 and the combination of these two (Figs. 4F and EV4G). Thus, the SGN pathway is essential for the shoot increase in anthocyanin production as a response to low soil nitrogen status. We investigated this further by fertilizing the CAS with nitrogen in the form of nitrate. Under these conditions, all parental lines and myb36-2 plants showed a strong growth promotion and no longer displayed excess anthocyanin accumulation (Fig. 5A–C). In line with the idea that this sensing was uncoupled, no significant growth promotion was observed in sgn3-3 or MYB36$_{Loop}$ in either WT or sgn3-3 background (Fig. 5A–C). Intriguingly, this effect was dependent on ligand-binding of CIF peptides to SGN3 as the cif1cif2 double mutant, showed a similar response as sgn3-3, while the sgn1-2 mutant showed a similar behavior as the parental plants despite its disturbed CS formation (Fig. 5C). After CIF activation of SGN3 a downstream response is phosphorylation and thus activation of the two NADP oxidases RBOHD and F, which leads to increased ROS formation (Fujita et al, 2020). We therefore investigated if the SGN3-dependent shoot nitrogen responses are facilitated by these enzymes. While rbohd mutants showed a similar response to the respective parental line, rbohf and rbohdf mutants had diminished nitrogen responses similar to those observed in MYB36$_{Loop}$ lines and sgn3-3 (Fig. 5B,C). Thus, we conclude that growth promotion of nitrogen under agricultural conditions is dependent on binding on CIF peptides to the SGN3 receptor and a

consecutive downstream activation of RBOHF. Combined, this suggests that the SGN pathway informs the shoot of soil nitrogen status by integrating CS formation into an RBOHF-dependent ROS signal (Fig. 6).

## Discussion

Positive feedback loops have been identified in genetic networks that control secondary cell wall formation (Taylor-Teeples et al, 2015) and play a role in reinforcing robust differential outputs in epidermal development (Kang et al, 2009). In such naturally occurring networks, the feedback regulation must be tightly controlled to avoid run-away expression. As this is not necessarily the case in an artificial version, this puts strain on the endogenous regulatory network. This can manifest either in the form of transcriptional co-factor depletion or compensatory repressive mechanisms such as silencing, that eventually must weaken the exponentially increasing expression encouraged by the artificial construct. If this is not the case, the plant will exhaust all resources in order to facilitate continuous expression. Thus, artificial positive feedback is an intriguing tool, which if not lethal, can be used to tease out which factors are rate limiting in a given genetic network. In the case of a MYB36-based loop system, we observed a clear activation, but also a relatively normal growth making this an intriguing tool for a deeper analysis of (epi)genetic silencing mechanisms that eventually shuts down the feedback mechanism. Combined, our data indicates that the MYB36$_{Loop}$ gives rise to earlier onset of CS and stronger deposition, consistent with previous observations of CS establishment where PI blockage was slightly delayed when compared with lignin autofluorescence in the endodermis (Naseer et al, 2012). However, with the abovementioned caveats in mind, further analysis into the mechanism(s) affected in these lines will reveal if the increased barrier function is related to increased barrier efficiency, affected deposition mechanisms and/or increased maturation of CS formation.

A number of studies have given comprehensive insights into the impact of root barriers on mineral accumulation in plant tissues, yet very little is known of barrier-associated changes in nitrogen-related responses. Our findings in an agar-based in vitro system revealed that the onset of a functional CS is dynamic and responsive to nitrogen (Fig. 3). An elegant way for the plant to integrate root growth responses into a perception mechanism of its barriers would be by linking nutrient response mechanisms directly to the degree of SGN signaling. This can either be by direct regulation of CS onset or more likely by changes in cell elongation,

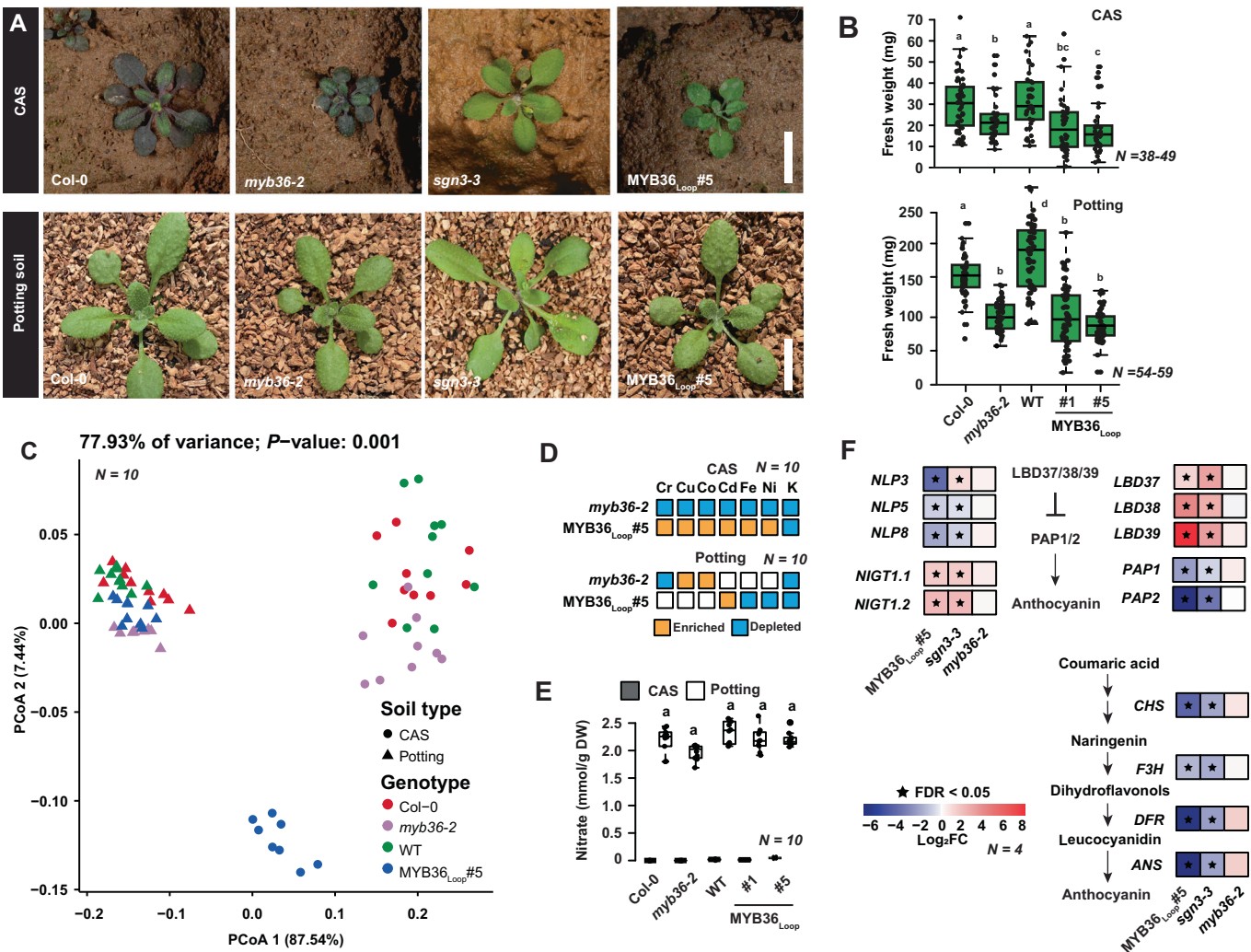

**Figure 4. Shoot analysis of plants with modified root barriers grown in different soil conditions.**

(A) Plants grown for four weeks on Cologne agricultural soil (CAS) or potting conditions (Potting). Scale bars: 1 cm. (B) Rosette fresh weight of 4-week-old plants grown on CAS (upper) or potting soil (lower). A representative results from two independent experiments. (C) PCoA plot of Bray–Curtis distances calculated on 24 shoot mineral contents of 4-week-old plants grown under CAS or standard potting soil conditions. 77.93% variation in shoot ionome can be explained by soil type ($P = 0.001$, PERMANOVA). (D) Minerals and (E) nitrate in rosettes grown on CAS or potting soil conditions. Minerals are normalized to the corresponding parental line (two-sided Student's $t$ test, $P < 0.05$). (F) Heatmap showing transcriptional behavior of NIN-LIKE-Proteins (NLPs), *NITRATE-INDUCIBLE, GARP-TYPE TRANSCRIPTIONAL REPRESSORs* (*NIGTs*), *BTB AND TAZ DOMAIN PROTEINs* (*BTs*) families, *PRODUCTION OF ANTHOCYANIN PIGMENT 1* and *2* (*PAP1* and *PAP2*) and anthocyanin synthesis genes under CAS condition. Stars represent significantly changed (FDR < 0.05) genes compared to parental line. For boxplots, center line in box indicates median, dots represent data, limits represent upper and lower quartiles, and whiskers maximum and minimum values. Letters depict statistical difference of one-way ANOVA with Holm–Sidak-adjusted post hoc $t$ test ($P < 0.05$). WT represents the parental line (pCASP1:CASP1-GFP) of MYB36$_{Loop}$ plants. Numbers of biological replicates are indicated on graph. Source data are available online for this figure.

which typically occur under nitrogen starvation (Kiba and Krapp, 2016). Our observation that the zone before functional CS onset was increased can be interpreted as a mechanism to increase the area where CIF peptides can diffuse from the stele before inhibition by functional CS establishment. One output from this would be a quantitative change in SGN3 activation, where a larger zone of diffusion translates into increased signaling, that via RBOHF-dependent ROS formation translocates to the shoots and serves as a nitrogen-status signal. This would allow the plant to fine-tune and coordinate CS status, nutrient uptake capacity and root growth responses via SGN3-dependent ROS formation. This is particularly

interesting in the context of our analysis on CAS, where such a system may serve to inform the shoot of the low nitrogen soil status (Fig. 6). In light of the proposed role of CS-SGN signaling in nitrogen-perception, the observed lack of nitrogen-induced growth promotion under CAS conditions fertilized with nitrate further implicates a distinct function in aiding the plant to utilize nitrate.

In summary, we here demonstrate that in an agriculturally relevant root environment, timing of the CS onset and the associated SGN system are employed by the plant as a mechanism to establish physiological homeostasis. This provides an updated insight into how plants integrate external soil inputs to their

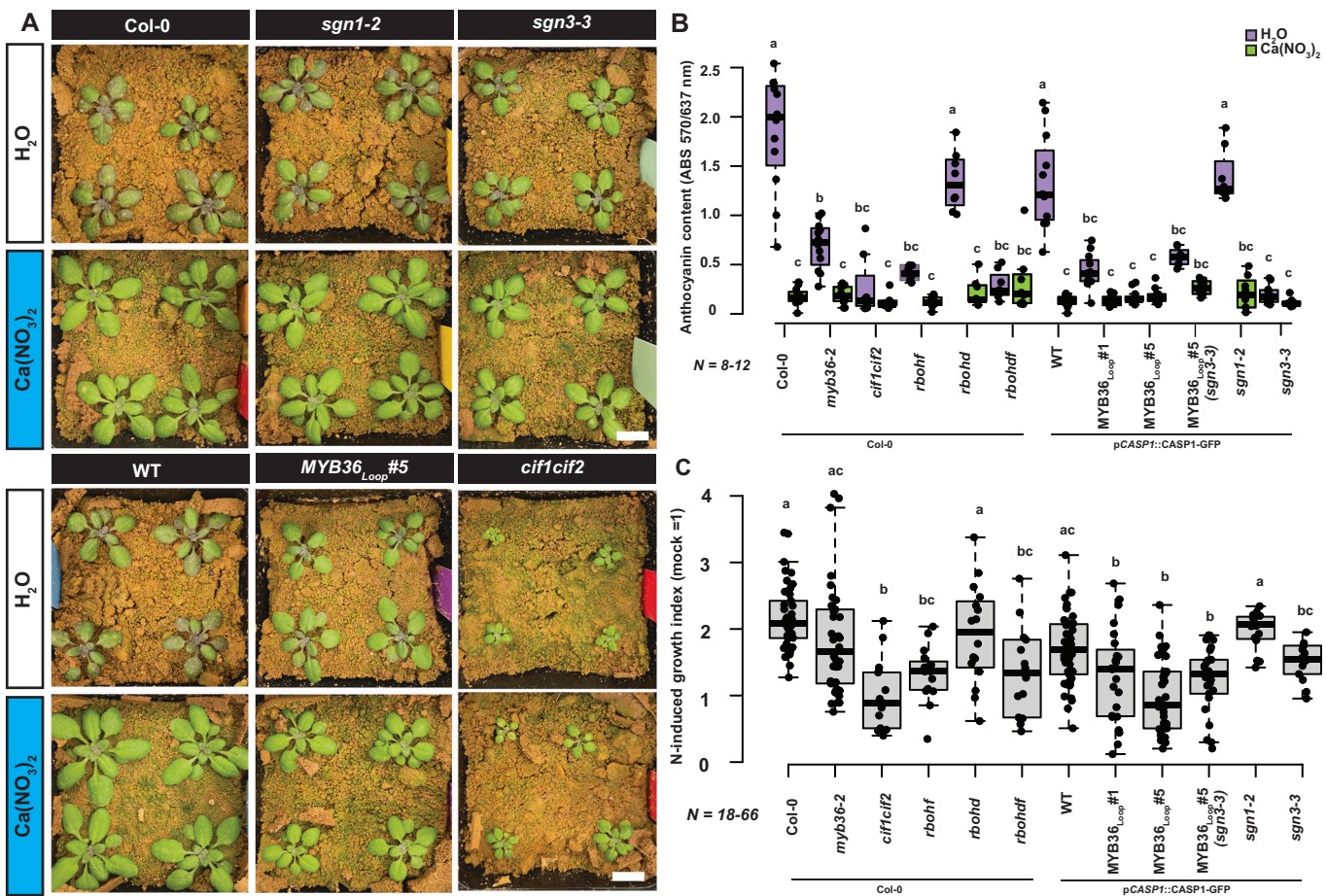

**Figure 5. Growth analysis of mutants grown on agricultural soil under a nitrogen fertilization scheme.**

(**A**) Four-week-old rosettes from different genotypes grown for one week on ½ MS conditions and transferred to Cologne agricultural soil (CAS) for three weeks and watered with water ($H_2O$) or a Calcinite™ solution containing nitrate ($Ca(NO_3)_2$). Scale bars represent 1 cm. (**B**) Anthocyanin content of 4-week-old rosettes from CAS watered with mock ($H_2O$) or nitrate ($Ca(NO_3)_2$). Representative result from two independent experiments. (**C**) Nitrogen-induced growth promotion (Calcinite/$H_2O$ weight) of 4-week-old rosettes from CAS. Representative result from two independent experiments. For boxplots, center line in box indicates median, dots represent data, limits represent upper and lower quartiles, and whiskers maximum and minimum values. Letters depict statistical difference of one-way ANOVA with Holm–Sidak-adjusted post hoc $t$ test ($P < 0.05$). WT represents the parental line (pCASP1:CASP1-GFP) of MYB36$_{Loop}$ plants. Numbers of biological replicates are indicated on graph. Source data are available online for this figure.

nutritional status and communication between the root and aboveground parts.

## Methods

### Plant growth

*Arabidopsis thaliana* ecotype Columbia-0 transgenic and mutant lines were used to perform experiments. Seeds were kept for 2 days at 4 °C in the dark for stratification. For nutrient starvation experiments plants were grown under 16 h light at 21 °C and 8 h dark at 19 °C vertically on ½ MS medium without sucrose but with distinct nutrient compositions. In all media, the pH was buffered to 5.8 using 10 mM 2-(N-morpholino) ethanesulfonic acid (MES). The following premixed media were used for nutrient starvation assays: Caisson MSP11, MSP44 or MSP21. Low nitrogen conditions were obtained by Caisson MSP21 + 0.037 mM KNO$_3$ + 0.037 mM

NH$_4$NO$_3$. For fresh weight measurements, shoots and roots of five plants were harvested after 14 days (3 days on ½ MS + 11 days on the specified condition). For Cologne agriculture soil (CAS) (Harbort et al, 2020) and potting soil (Blumavis Mini Tray MIM800) experiments, 4–5 plants were grown in 9 × 9 cm square pots in a climate-controlled greenhouse, at 21 °C under a 16 h light/8 h dark regime. For CAS experiments pots were placed on top of a capillary mattress in a tray, and tap water or calcinit™ solution (0.651 g/L, calcinit™ composition: 14.4% nitrate, 1% ammonium, 26% calcium oxide) (Yara, Germany) was added. For potting soil experiments, soil was moisturized by submerging the pot bottom in water regularly. Plants were randomized for all experiments, but measurements were performed without blinding.

### Cloning

To generate the endodermis-specific feedback loop expression constructs, the coding sequence of MYB36 (Kamiya et al, 2015) was

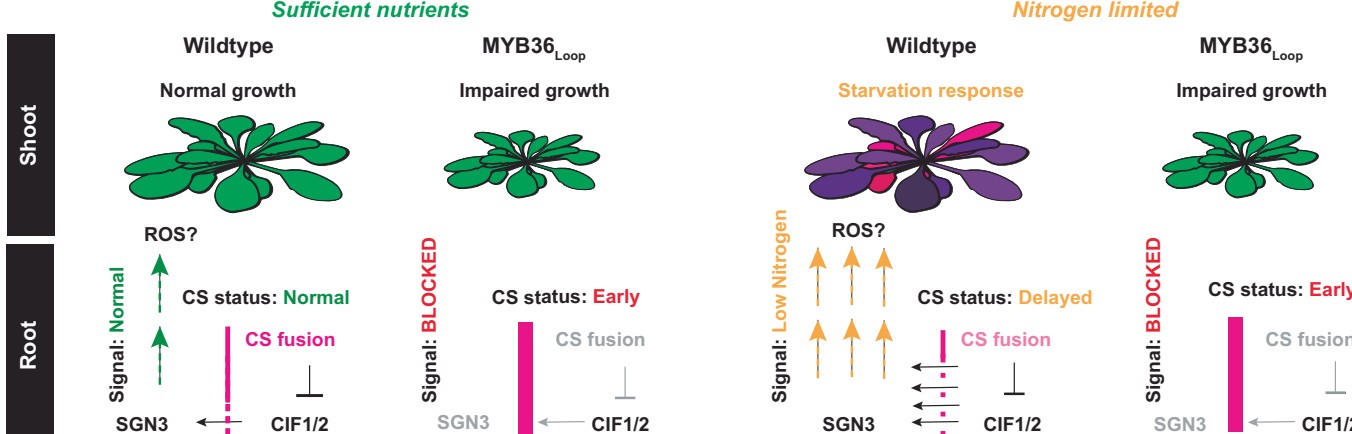

**Figure 6. Model for Casparian strip-related signaling across different soil conditions.**

In roots Casparian strip (CS) fusion depends on activation of SGN3 via diffusion of the SGN3 ligands known as Casparian strip integrity factor (CIF) peptides from the stele across the endodermis. In our model, this feature additionally provides information to the shoot. Upon low nitrogen status in the soil, the CS establishment is delayed and gives rise to an increased diffusion of CIF and thereby a quantitative difference in SGN signaling. The output of this modifies downstream responses, which ultimately affects the shoots ability to sense soil status. Upon increased CS formation in the MYB36$_{Loop}$ lines, this signaling pathway is overridden, which blocks activation of the SGN pathway and makes the shoot unable to sense the soil environment. This is likely due to a disturbed ability of the root to facilitate long-distance signaling receptor through reactive oxygen species (ROS) produced by the NADP oxidase RBOHF. In case of disturbed CS formation (i.e., *myb36* KO) the lack of CS induces an ABA-dependent SOS response in roots which is propagated independently to the shoot and capable of compensation responses.

Gateway-cloned into a pDONR221 entry vector using BP clonase II (Invitrogen) according to the manufacturer's description. Together with previously generated P4L1r pDONR entry vectors containing the pCASP1 sequence (Roppolo et al, 2011), this was recombined using LR-clonase II (Invitrogen) into a FastRed selection-containing destination vector (pED97) (Andersen et al, 2018). The final construct was transformed into *pCASP1*::CASP1-GFP and other backgrounds using the floral dip method and selected using FastRed selection.

## Staining procedures

All staining procedures were done using ClearSee staining (Kurihara et al, 2015; Ursache et al, 2018). Briefly, plants were fixed in 3 mL 1× PBS containing 4% p-formaldehyde for 1 h at room temperature and washed twice with 3 mL 1× PBS. Following fixation, the seedlings were cleared in 3 mL ClearSee solution (10% xylitol, 15% sodium deoxycholate, and 25% urea in water) under gentle shaking. After overnight clearing, the solution was exchanged to a new ClearSee solution containing 0.2% Basic Fuchsin and 0.1% Calcofluor White for lignin and cell wall staining, respectively. The dye solution was removed after overnight staining and rinsed once with fresh ClearSee solution. The samples were washed for 30 min with gentle shaking followed by overnight incubation in ClearSee solution before imaging. Suberin staining of Arabidopsis roots was performed as previously described (Ursache et al, 2018). In case of combined suberin and lignin staining the procedure was according to (Sexauer et al, 2021). Briefly, vertically grown 5-day-old seedlings were incubated solution of Fluorol Yellow 088 (Sigma) (0.01%, in lactic acid) and incubated for 30 min at 70 °C. The stained seedlings were rinsed shortly in water and transferred to a freshly prepared solution of Aniline blue (0.5%, in water) for counterstaining. Propidium iodide (PI) assays were done

as described (Naseer et al, 2012). PI staining seedlings were washed for 2–3 min in water and transferred to a chambered cover glass (Thermo Scientific), and imaged either using Confocal laser scanning (CLSM) microscopy or epifluorescence microscopy.

## Microscopy

All confocal images were taken using a Zeiss LSM 980 system. Fluorophore settings were ex 488 nm, em 505–550 nm for GFP and Fluorol Yellow 088, ex 594 nm, em 610–650 nm for mCherry, ex 561 nm, em 580–600 nm for Basic fuchsin, ex 405 nm em 420–430 nm for Calcofluor White. Airyscan images were taken using the 4Y multiplex setting. FY staining and PI analysis were done on a Zeiss Axiozoom V16 system (GFP filtercube ex: 470 nm/40 em:525/50 bs: 500). For PI TX2 filtercube ex: 560 nm/40 em:645/75 bs: 595 nm).

## Transmission electron microscopy

For quantification of the Casparian strip width by TEM, 6-day-old pCASP1::CASP1-GFP and MYB36$_{Loop}$#5 seedlings were placed in small Petri dishes filled with 0.05 M MOPS buffer supplemented with 0.1% Tween20 and 10 mM cerium chloride and incubated with gentle agitation at room temperature. After 30 min, the CeCl$_3$-containing MOPS buffer was replaced by 2.5% glutaraldehyde in 0.05 M phosphate buffer (v/v), pH 7.2, and seedlings were gently agitated for one hour. Subsequently, seedlings were transferred into glass vials filled with 1% osmium tetroxide (EMS, #19150) in 0.05 M phosphate buffer (w/v), pH 7.2, supplemented with 1.5% potassium ferrocyanide, for another hour. After three rinses in water, seedlings were embedded into 2% low melting temperature agarose and three fragments were sampled from the distal 1.5 cm-long part of the root. Root fragments were then dehydrated with a

series of ethanol, gradually transferred into acetone, and embedded into Araldite 502/Embed 812 resin (EMS, #13940) using the EMS Poly III embedding machine (EMS, #4444). Ultrathin sections ($\approx$70 nm) were cut at specific distances from the root tip with a Reichert-Jung Ultracut E and collected on Formvar-coated copper slot grids (Moran and Rowley, 1987). After staining with 0.1% potassium permanganate in 0.1 N $H_2SO_4$ for one minute (w/v) followed by 0.5% uranyl acetate in water (w/v) for 10 min and lead citrate for 15 min, sections were examined with a Hitachi H-7650 TEM operating at 100 kV and equipped with an AMT XR41-M digital camera. For immunogold detection of CASP1-GFP in MYB36$_{Loop}$ roots were high-pressure frozen in a Leica EM HPM 100 high-pressure freezer between two large aluminum specimen carriers (ø 4.6 mm) enclosing a cavity of 150 µm depth filled with ½ MS medium. After freeze substitution in the Leica EM AFS2 freeze substitution device using 0.5% uranyl acetate in acetone (w/v), bringing samples from −85 °C to −20 °C over seven days, samples were transferred to ethanol and gradually embedded into LR White resin (Plano GmbH, R1281) at −20 °C over 6 days with constant agitation. Samples were polymerized in pure LR White resin with UV light for 24 h at −20 °C and 24 h at 0 °C. Ultramicrotomy was performed as described above with the exception that sections were collected on Formvar-coated gold slot grids. Immunogold labeling of CASP1-GFP was carried out according to (Viñegra de la Torre et al, 2022) using a 1:5 dilution of rat monoclonal anti-GFP 3H9 (Chromotek) and a 1:20 dilution of goat anti-rat IgG conjugated to 10-nm colloidal gold particles (British Biocell International). Sections were stained with potassium permanganate and uranyl acetate (no lead citrate) and imaged as described above.

## Transcriptomics

For transcriptomic analysis on roots, Col-0, *myb36-2*, *pCASP1::-CASP1-GFP* (WT), MYB36$_{Loop}$#1 and MYB36$_{Loop}$#5 seedlings were grown on standard ½ MS agar medium for seven days. Whole roots were harvested and immediately frozen in liquid nitrogen. RNA was extracted using a TRIzol (Invitrogen)-adapted ReliaPrep RNA extraction kit (Promega) (Andersen et al, 2018). For transcriptomic analysis on CAS and potting soil grown shoots (set 1), Col-0, *myb36-2*, WT, and MYB36$_{Loop}$#5 seedlings were grown on standard solid ½ MS medium for 7 days, then transferred to CAS soil or potting soil. Note that transcriptomic analysis on CAS and potting soil grown *sgn3-3* shoots was an independent experiment (set 2), WT and *sgn3-3* (expressing *pCASP1::CASP1-GFP*) samples were prepared as mentioned above. Whole rosette leaves of 4-week-old plants were harvested and immediately frozen in liquid nitrogen. RNA extractions were performed as mentioned above. RNA quality was determined using a Bioanalyzer 2100 system (Agilent Technologies, USA). Library preparation and paired-end 150 bp sequencing were conducted by Novogene (Cambridge, UK). Approximately 40 million raw reads were generated per sample. RNA-seq raw reads of *sgn3-3* roots were acquired from (Fujita et al, 2020; Reyt et al, 2021), raw reads of CIF2 treated Col-0 roots dataset (48 h after CIF2 treatment) were acquired from (Fujita et al, 2020). All raw reads were preprocessed using fastp (v0.22.0) (Chen et al, 2018). Filtered high-quality reads were mapped to *A. thaliana* TAIR10 reference genome with Araport 11 annotation (Phytozome genome ID: 447) using HISAT2 (v2.2.1), and counted using

featureCounts from the Subread package (v2.0.1) (Chen et al, 2018). All statistical analyses were performed using R (v4.1.2) (https://www.R-project.org/). Read counts were transformed to cpm (counts per million) using edgeR package (Robinson et al, 2009). Lowly expressed (less than 0.5× total sample number 19 for ½ MS roots, 26 for CAS shoots set 1, 8 for CAS shoots set 2, 16 for potting shoots set 1, 8 for potting shoots set 2, 6 for (Reyt et al, 2021) 9 for (Fujita et al, 2020)) cpm over all samples, and at least 1 cpm in n samples (6 for ½ MS roots, 8 for CAS shoots set 1, 2 for CAS shoots set 2, 5 for potting shoots set 1, 2 for potting shoots set 2, 2 for (Reyt et al, 2021), 3 for (Fujita et al, 2020)) genes were removed from the analysis. Differentially expressed genes (DEGs) were identified by pairwise comparisons using the glmFit function in edgeR package (Robinson et al, 2009) with absolute fold change more than 2 and a false discovery rate (FDR) corrected *P* value less than 0.05. Note that for root transcriptome analysis, MYB36$_{Loop}$#1 and MYB36$_{Loop}$#5 samples were compiled as one genotype, and compared with WT. Gene Ontology (GO) term enrichment analysis was performed using Metascape (Zhou et al, 2019). Only GO terms with adjusted *P* value (*q*-value) lower than 0.05 were considered as significantly enriched. Heatmaps were generated using pheatmap package (https://cran.r-project.org/web/packages/pheatmap/index.html) or ggplot2 package (Wickham, 2016). Upset plots were generated using the UpSetR package.

## ICP-MS and IC analysis

The total mineral content was determined by inductively-coupled plasma mass spectrometry (ICP-MS) following the method described in (Almario et al, 2017). Approximately 5 mg of homogenized dried plant material were digested using 500 µL of 67% (w/w) $HNO_3$ overnight at room temperature and subsequently placed in a 95 °C water bath for 30 min or until the liquid was completely clear. After cooling to room temperature, the samples were placed on ice and 4.5 mL of deionized water was carefully added to the tubes. The samples were centrifuged at 4 °C at $2000 \times g$ for 30 min and the supernatants were transferred to new tubes. The elemental concentration was determined using Agilent 7700 ICP-MS (Agilent Technologies) (Almario et al, 2017). Inorganic anion (nitrate, phosphate, and sulfate) levels were measured by ion chromatography, as described in (Dietzen et al, 2020). Approximately 10 mg of dried plant material was homogenized in 1 mL deionized water, shaken for 1 h at 4 °C, and subsequently heated at 95 °C for 15 min. The anions were determined by the Dionex ICS-1100 chromatography system and separated on a Dionex IonPac AS22 RFIC 4× 250 mm analytic column (Thermo Scientific, Darmstadt, Germany), using 4.5 mM $Na_2CO_3$/1.4 mM $NaHCO_3$ as running buffer (Dietzen et al, 2020). To compare the shoot ionomes between different genotypes under different growth conditions, a principal coordinate analysis (PCoA) was performed with a Bray–Curtis dissimilarity index calculated using vegdist() function in the R vegan package (https://github.com/jarioksa/vegan). To assess the variations explained by growth conditions (CAS vs. potting soil), a PERMANOVA testing was performed, based on the Bray–Curtis dissimilarity index, using adonis2() function from the R vegan package with 999 permutations. To assess the statistical difference of shoot ionomes between different genotypes within each growth condition, a pairwise comparison (PERMANOVA) was performed, based on the Bray–Curtis

dissimilarity index, using the pairwise.adonis() function in the R pairwise Adonis package (https://github.com/pmartinezarbizu/pairwiseAdonis) with 999 permutations, followed by Benjamini–Hochberg post hoc testing.

## Quantitative RT-PCR

For qRT-PCR, samples were prepared as abovementioned. Each biological replicate represents an individual whole rosette. RNA extraction was performed using a TRIzol (Invitrogen)-adapted ReliaPrep RNA extraction kit (Promega), as abovementioned. cDNA synthesis was performed using iScript™ cDNA Synthesis Kit (BioRad) in a final volume of 20 µL. Each reaction contained 1 µg of total RNA. qRT-PCR was performed in a BioRad CFX Connect Real-Time system in a final volume of 10 µL. Each reaction contained 5 µL of 2× iQ SYBR Green supermix (BioRad), 2 µL of diluted cDNA (10 times dilution), 1 µL of 2.5 µM forward primer, 1 µL of 2.5 µM reverse primer and 1 µL of water. *EF1* (AT1G07920) was used as the reference. The thermal cycler conditions were: 95 °C for 2 min, 40 cycles of 95 °C for 30 s, 60 °C for 30 s, and 72 °C for 30 s. For the melting curve, conditions were set as: denaturation, 95 °C for 10 s; hybridization, 60 °C for 5 s; denaturation until 95 °C with 0.5 °C incrementation. Relative expression values were determined using the $2^{-\Delta\Delta CT}$ method. qRT-PCR primers used in this study are listed in Dataset EV1.

## Anthocyanin content measurement

Measurement of anthocyanin content in rosettes grown under CAS or standard potting soil conditions was performed as previously described (Nakata and Ohme-Takagi, 2014). Absorbances at 530 and 637 nm were measured using Tecan Infinite 200 PRO plate reader.

## Data availability

RNA-seq raw reads generated in this study have been deposited at the National Center for Biotechnology Information under BioProject ID PRJNA940103.

The source data of this paper are collected in the following database record: biostudies:S-SCDT-10_1038-S44318-024-00107-3.

## Peer review information

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

## Acknowledgements

All authors thank Bart Boesten, and Ila Rouhara for technical support. Sabine Ambrosius and the Biocenter MS Platform Cologne for measurements of mineral composition. Ton Timmers and the Central Microscopy facility (CeMic) are thanked for microscopy aid. Aristeidis Stamatakis and his greenhouse team at MPIPZ are thanked for help with plant growth. We moreover thank Meike Burow, Niko Geldner, Magdalena Marek, Sebastian Samwald, Lioba Rüger and Marc Somssich for insightful comments on the manuscript. Joop Vermeer and his lab are thanked for their comments on our preprint version of this work. We also thank Satoshi Fujita and Hiroko Uchida for graphical support. TGA would further like to thank BDR and APM for their contribution of social media at timed intervals. Research in the lab of TGA is supported by the Sofja Kovalevskaja programme from the Alexander von Humboldt foundation and the Max Planck Society. Research in the lab of SK is supported by the Deutsche Forschungsgemeinschaft (DFG) under Germany´s Excellence Strategy—EXC 2048/1— project 390686111. KW is funded by DFG Priority Programme SPP2125 DECRyPT.

## Author contributions

**Defeng Shen**: Conceptualization; Data curation; Software; Formal analysis; Validation; Investigation; Visualization; Methodology; Writing—original draft; Writing—review and editing. **Kathrin Wippel**: Conceptualization; Formal analysis; Investigation; Methodology. **Simone Remmel**: Data curation; Investigation. **Yuanyuan Zhang**: Formal analysis; Investigation; Methodology. **Noah Kuertoes**: Data curation; Formal analysis; Investigation; Methodology. **Ulla Neumann**: Data curation; Formal analysis; Investigation; Methodology. **Stanislav Kopriva**: Data curation; Funding acquisition; Investigation. **Tonni**

**Grube Andersen**: Conceptualization; Supervision; Funding acquisition; Investigation; Visualization; Writing—original draft; Project administration; Writing—review and editing.

Source data underlying figure panels in this paper may have individual authorship assigned. Where available, figure panel/source data authorship is listed in the following database record: biostudies:S-SCDT-10_1038-S44318-024-00107-3.

## Funding

## Disclosure and competing interests statement
The authors declare no competing interests.

# Expanded View Figures

**Figure EV1.  Detailed analysis of Casparian strip formation and suberization in mutants affected in Casparian strip formation.**

(A) Upset plot showing the number of differentially expressed genes (DEGs) in MYB36$_{Loop}$ and *myb36-2* roots grown on standard ½ MS agar medium, comparing with the DEGs in Col-0 roots treated with CIF2 from (Fujita et al, 2020). Orange bars represent genes upregulated in MYB36$_{Loop}$, downregulated in *myb36-2*; blue bars represent genes downregulated in MYB36$_{Loop}$, upregulated in *myb36-2*. Red rectangle highlights Schengen pathway activated genes, with GO terms enriched. (B) Gene Ontology (GO) term enrichment of DEGs. Note that GO terms enriched in MYB36 $_{Loop}$ downregulated genes are largely overlapping with the GO terms enriched in Schengen pathway activated genes in (A). Gene ratio represents the number of DEGs of GO term divided by the total genes in the GO term. + represents upregulated genes, - represents downregulated genes. (C) Venn diagram showing the overlap of downregulated genes in MYB36$_{Loop}$roots, downregulated genes in *sgn3-3* roots from (Reyt et al, 2021) and downregulated genes in Col-0 roots treated with CIF2 (Fujita et al, 2020). (D) Bar plots showing the distance between root tip and start of string-of-pearls CSD region (left graph) and the distance between root tip and start of xylem (right graph). Plants were grown on standard ½ MS agar medium for 8 days. Data are mean ± SD. Statistical significance of differences with the parental line (WT) was determined using a two-tailed Student's t test. ns; not significant. (E) Transmission electron microscopy (TEM) micrograph of 7-day-old anti-GFP immunogold-labeled MYB36$_{Loop}$ #5 roots. Scale bar represents 500 nm. Co; Cortex, CS; Casparian strip, En: Endodermis. (F) Maximum projection of a confocal image stack of 7-day-old MYB36$_{Loop}$ plants expressing p*PER64*::PER64-mCherry and p*CASP1*::CASP1-GFP. Scale bars represent 10 μm. Line in overlay depicts the transect used for relative intensity measurements. (G) Endodermal cells of plants expressing MYB36$_{Loop}$ stained with Basic fuchsin and Fluorol yellow. Scale bars represent 5 μm. (H) Measurement of suberization pattern under mock or 100 nM CIF2 conditions. Error bar represents standard deviation. (I) Top-view maximum projection of 7-days old MYB36$_{Loop}$#5 plants treated with 100 nM CIF2 for 24 h before fixing in Clearsee and imaged using a confocal setup. Scale bars represent 2 μm. Different letters depict statistical difference in a one-way ANOVA analysis with Tukey's test ($P < 0.05$). WT represents the parental line (pCASP1:CASP1-GFP) of MYB36$_{Loop}$ plants. Numbers of biological replicates are indicated on graph.

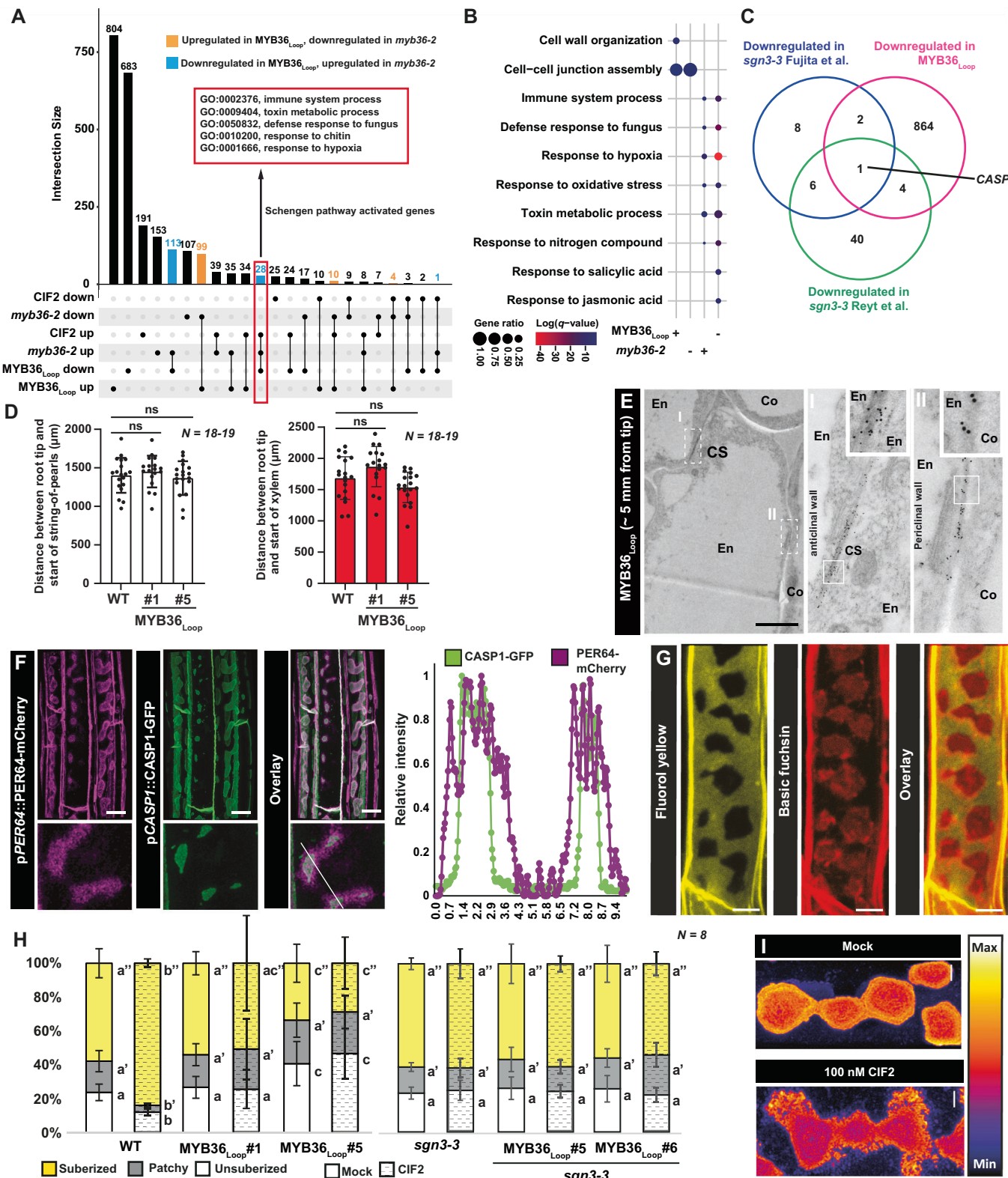

**A** Upregulated in MYB36_Loop, downregulated in myb36-2 / Downregulated in MYB36_Loop, upregulated in myb36-2

GO:0002376, immune system process
GO:0009404, toxin metabolic process
GO:0050832, defense response to fungus
GO:0010200, response to chitin
GO:0001666, response to hypoxia

Schengen pathway activated genes

**B** Cell wall organization / Cell−cell junction assembly / Immune system process / Defense response to fungus / Response to hypoxia / Response to oxidative stress / Toxin metabolic process / Response to nitrogen compound / Response to salicylic acid / Response to jasmonic acid

Gene ratio / Log(q−value)

**C** Downregulated in sgn3-3 Fujita et al. / Downregulated in MYB36_Loop / Downregulated in sgn3-3 Reyt et al. / CASP4

**D** Distance between root tip and start of string-of-pearls (μm) / Distance between root tip and start of xylem (μm) / MYB36_Loop

**E** MYB36_Loop (~ 5 mm from tip) / En / Co / CS / anticlinal wall / Periclinal wall

**F** pPER64::PER64-mCherry / pCASP1::CASP1-GFP / Overlay / CASP1-GFP / PER64-mCherry / Relative intensity

**G** Fluorol yellow / Basic fuchsin / Overlay

**H** Suberized / Patchy / Unsuberized / Mock / CIF2

**I** Mock / 100 nM CIF2 / Max / Min

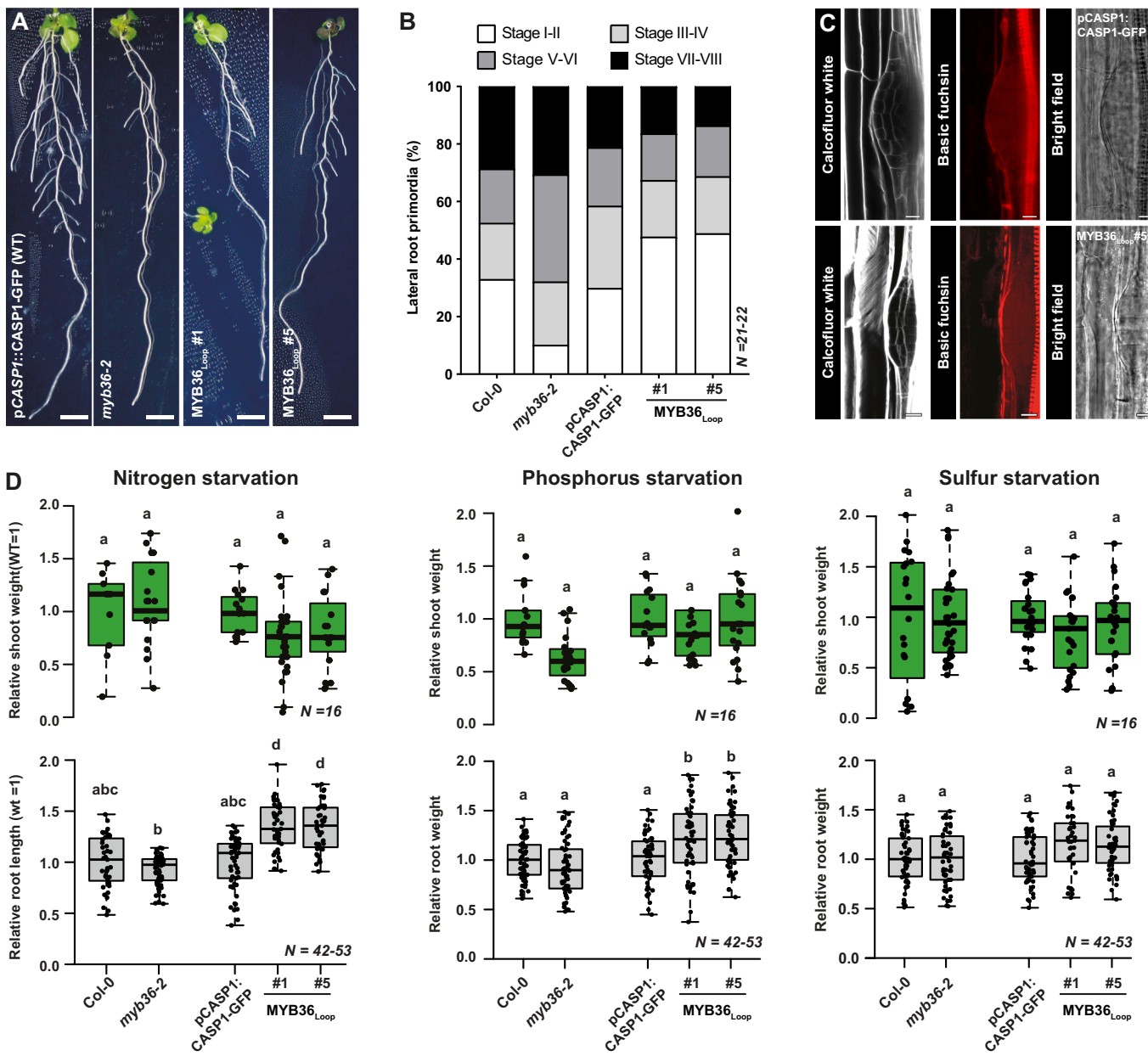

**Figure EV2.  Detailed physiological and anatomical analysis of mutants affected in Casparian strip formation.**

(**A**) 14-day-old seedlings grown on standard ½ MS agar medium. Scale bars represent 5 mm. (**B**) The distribution of different stages of Lateral root primordia (LRP) of 8-day-old plants grown on standard ½ MS agar medium. Combined data from two independent experiments. (**C**) LRP of 8-day-old Col-0 (upper graph) and MYB36$_{Loop}$#5 (lower graph) roots grown on standard ½ MS agar medium, stained with cell wall dye Calcofluor White and the lignin-specific dye Basic Fuchsin. Scale bars represent 10 μm. (**D**) Measurement of shoot (upper graph) and root (lower graph) fresh weight of 2-week-old plants grown under nitrogen, phosphorous (left) or sulfur (right) starvation conditions. The weight was normalized to the changes in the corresponding parental background (Col-0 for *myb36-2* and pCASP1::CASP1-GFP for MYB36$_{Loop}$ lines). For boxplots, the center line in the box indicates the median, dots represent data, the box limits represent the upper and lower quartiles, and the whiskers represent the maximum and minimum values. Different letters depict statistical difference in a one-way ANOVA analysis with Tukey's test (*P* < 0.05). Numbers of biological replicates are indicated on graph.

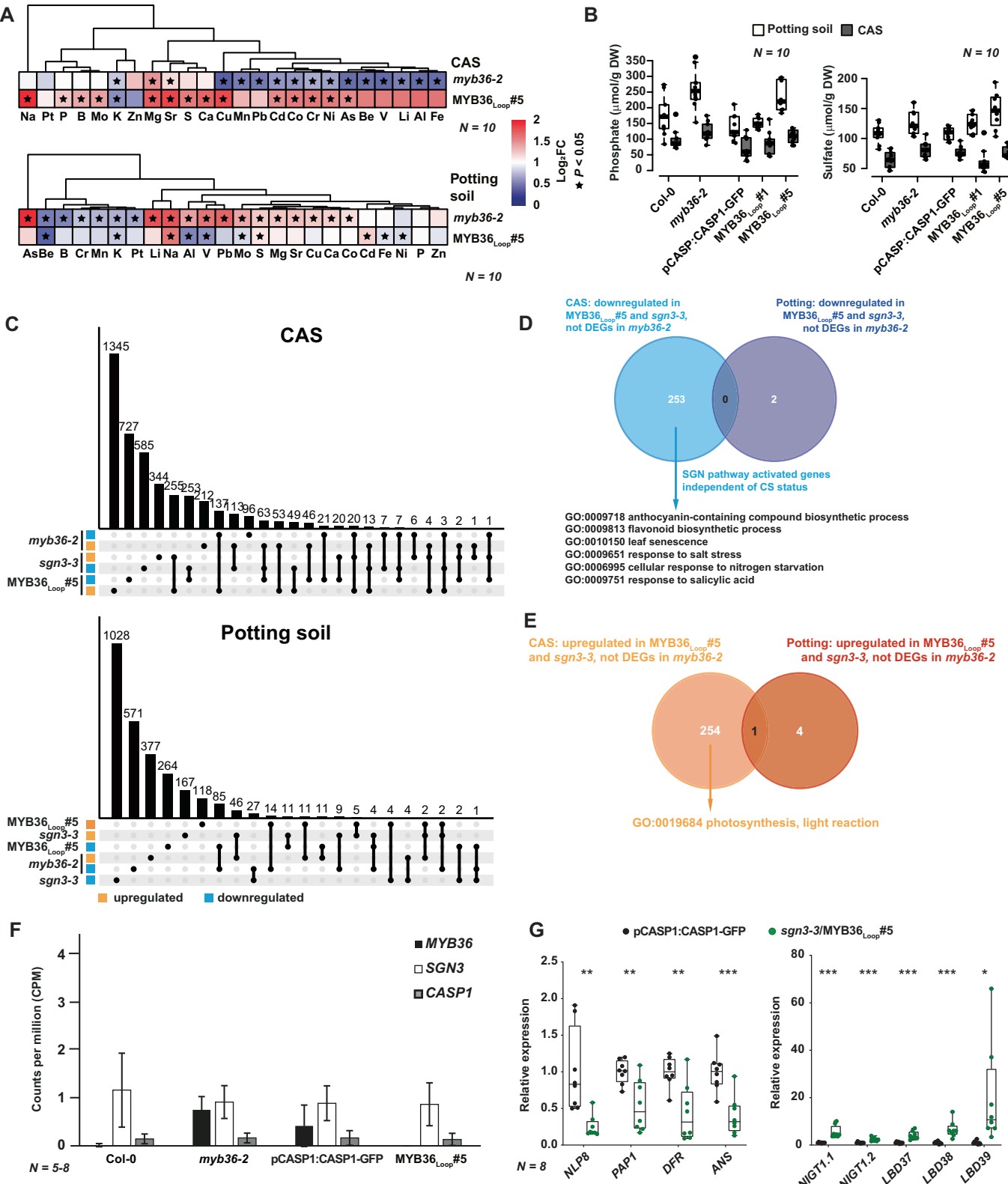

**Figure EV3.   Detailed transcript- and ionome analysis of soil-grown plants affected in Casparian strip formation.**

(A) Heatmap of 24 element contents in 4-week-old MYB36$_{Loop}$#5 and *myb36*-2 rosettes grown under CAS (upper) or standard potting soil (lower) conditions after normalization to the corresponding parental lines. Star symbols represent significant differences with respective wild-type lines determined using a two-tailed Student's t test ($P < 0.05$). (B) Measurement of phosphate (left) and sulfate (right) content in 4-week-old rosettes grown under CAS or standard potting soil conditions. (C) Upset plots showing the number of DEGs in MYB36$_{Loop}$ #5, *sgn3-3* and *myb36-2* rosettes grown under CAS (upper) or standard potting soil (lower) conditions. Orange: upregulated; blue: downregulated. (D) Venn diagram depicting the overlap of DEGs in rosettes grown under CAS and standard potting conditions and the GO terms specifically enriched among the 254 upregulated DEGs in MYB36$_{Loop}$#5 and *sgn3-3* rosettes, but not DEGs in *myb36-2* rosettes under CAS condition. (E) Venn diagram depicting the overlap of upregulated DEGs in *myb36*-2, but not DEGs in *sgn3-3* rosettes between CAS and standard potting condition, and the GO terms enriched. The overlap represents 19 genes activated by the Schengen (SGN) pathway due to defective Casparian strip (CS) formation in both soil conditions. (F) Bar plot indicating expression of *MYB36*, *SGN3* and *CASP1* in rosettes of CAS-grown plants. Bars indicate mean values, error bars represent standard deviation. (G) RT-qPCR analysis of gene expression level in *sgn3-3*/MYB36$_{Loop}$#5 rosettes compared with the parental line pCASP1::CASP1-GFP grown on CAS with water. The center line in the box indicates the median, dots represent data, the box limits represent the upper and lower quartiles, and the whiskers represent the maximum and minimum values. Combined data from two independent experiments, analyzed by Student's *t* test (*$P < 0.05$, **$P < 0.01$, ***$P < 0.001$). Numbers of biological replicates are indicated on graph.

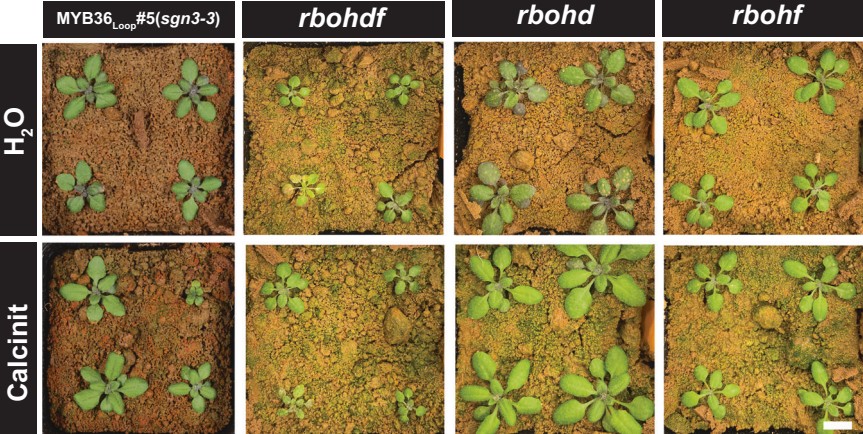

**Figure EV4. Shoot phenotypes of plants grown under agricultural conditions.**

Four-week-old rosettes from different genotypes grown for one week on ½ MS conditions and transferred to Cologne agricultural soil (CAS) for 3 weeks and watered with water ($H_2O$) or a Calcinite™ solution containing nitrate ($Ca(NO_3)_2$). Scale bars represent 1 cm. Note that the images of *sgn3-3*/MYB36$_{Loop\#5}$ were derived from an independent experiment from the rest.

