## [Peer Review File · The EMBO Journal]

The *Arabidopsis* SGN3/GSO1 receptor kinase integrates soil nitrogen status into shoot development

Tonni Andersen, Defeng Shen, Kathrin Wippel, Simone Rimmel, Yuanyuan Zhang, Noah Kuertoos, Ulla Neumann, and Stanislav Kopriva

Corresponding author(s): Tonni Andersen (tandersen@mpipz.mpg.de)

Review Timeline:

Submission Date:	20th Nov 23
Editorial Decision:	25th Jan 24
Revision Received:	2nd Mar 24
Editorial Decision:	14th Mar 24
Revision Received:	18th Mar 24
Accepted:	4th Apr 24

Editor: William Teale

Transaction Report:

Dear Tonni,

Thank you again for sharing your work and the submission of your manuscript entitled 'The SGN3/GSO1 receptor kinase integrates soil nitrogen status with shoot responses' (EMBOJ-2023-116190) to The EMBO Journal. Please accept my sincere apologies for the unusually long peer-review period take for your study. Your manuscript was sent to three reviewers for evaluation; we have received reports from two of them, which I enclose below. Please note that while feedback from referee #3 is still pending at this stage we have, in light of the other reviewers' input, decided to proceed with our decision in order to expedite the manuscript's processing.

As you will see, both referees acknowledge the potential interest and novel approach taken by your work, as well as the level of robustness and clarity needed for publication in The EMBO Journal.

I judge the comments of the referees to be generally reasonable and given their overall interest, we are in principle happy to invite you to revise your manuscript to address the referees' comments. This decision must be, however, contingent on there being no technically overriding concerns presented by referee #3 (whose comments I will share as soon as I receive them). In this revision, please pay particular attention to the referees' comments about the genetic backgrounds in which you make your phenotypic comparisons. This aspect of the manuscript needs further development and clearer reporting.

I should add that it is The EMBO Journal policy to allow only a single major round of revision and that it is therefore important to resolve the main concerns at this stage. I believe the concerns of the referees are reasonable and addressable, but please contact me if you have any questions, need further input on the referee comments or if you anticipate any problems in addressing any of their points. Please, follow the instructions below when preparing your manuscript for resubmission.

I would also like to point out that as a matter of policy, competing manuscripts published during this period will not be taken into consideration in our assessment of the novelty presented by your study ("scooping" protection). We have extended this 'scooping protection policy' beyond the usual 3 month revision timeline to cover the period required for a full revision to address the essential experimental issues. Please contact me if you see a paper with related content published elsewhere to discuss the appropriate course of action.

Again, please contact me at any time during revision if you need any help or have further questions.

Thank you very much again for the opportunity to consider your work for publication. I look forward to your revision.

Best regards,

William

William Teale, Ph.D.
Editor
The EMBO Journal

When submitting your revised manuscript, please carefully review the instructions below and include the following items:

- 1) a .docx formatted version of the manuscript text (including legends for main figures, EV figures and tables). Please make sure that the changes are highlighted to be clearly visible.
- 2) individual production quality figure files as .eps, .tif, .jpg (one file per figure).
- 3) a .docx formatted letter INCLUDING the reviewers' reports and your detailed point-by-point response to their comments. As part of the EMBO Press transparent editorial process, the point-by-point response is part of the Review Process File (RPF), which will be published alongside your paper.
- 4) a complete author checklist, which you can download from our author guidelines ([https://wol-prod-cdn.literatumonline.com/pb-assets/embo-site/Author Checklist%20-%20EMBO%20J-1561436015657.xlsx](https://wol-prod-cdn.literatumonline.com/pb-assets/embo-site/Author%20Checklist%20-%20EMBO%20J-1561436015657.xlsx)). Please insert information in the checklist that is also reflected in the manuscript. The completed author checklist will also be part of the RPF.

6) We require a 'Data Availability' section after the Materials and Methods. Before submitting your revision, primary datasets produced in this study need to be deposited in an appropriate public database, and the accession numbers and database listed under 'Data Availability'. Please remember to provide a reviewer password if the datasets are not yet public (see <https://www.embopress.org/page/journal/14602075/authorguide#datadeposition>). If no data deposition in external databases is needed for this paper, please then state in this section: This study includes no data deposited in external repositories. Note that the Data Availability Section is restricted to new primary data that are part of this study.

Note - All links should resolve to a page where the data can be accessed.

8) For data quantification: please specify the name of the statistical test used to generate error bars and P values, the number (n) of independent experiments (specify technical or biological replicates) underlying each data point and the test used to calculate p-values in each figure legend. The figure legends should contain a basic description of n, P and the test applied. Graphs must include a description of the bars and the error bars (s.d., s.e.m.).

9) We would also encourage you to include the source data for figure panels that show essential data. Numerical data can be provided as individual .xls or .csv files (including a tab describing the data). For 'blots' or microscopy, uncropped images should be submitted (using a zip archive or a single pdf per main figure if multiple images need to be supplied for one panel). Additional information on source data and instruction on how to label the files are available at .

10) We replaced Supplementary Information with Expanded View (EV) Figures and Tables that are collapsible/expandable online (see examples in <https://www.embopress.org/doi/10.15252/embj.201695874>). A maximum of 5 EV Figures can be typeset. EV Figures should be cited as 'Figure EV1, Figure EV2" etc. in the text and their respective legends should be included in the main text after the legends of regular figures.

12) Our journal encourages inclusion of *data citations in the reference list* to directly cite datasets that were re-used and obtained from public databases. Data citations in the article text are distinct from normal bibliographical citations and should directly link to the database records from which the data can be accessed. In the main text, data citations are formatted as follows: "Data ref: Smith et al, 2001" or "Data ref: NCBI Sequence Read Archive PRJNA342805, 2017". In the Reference list, data citations must be labeled with "[DATASET]". A data reference must provide the database name, accession number/identifiers and a resolvable link to the landing page from which the data can be accessed at the end of the reference. Further instructions are available at .

Further instructions for preparing your revised manuscript:

We realize that it is difficult to revise to a specific deadline. In the interest of protecting the conceptual advance provided by the work, we recommend a revision within 3 months (24th Apr 2024). Please discuss the revision progress ahead of this time with the editor if you require more time to complete the revisions. Use the link below to submit your revision:

Referee #1:

Shen et al. Review

In this manuscript, Shen et al. investigate the Schengen (SGN) pathway, a receptor-ligand system so far mainly associated with ensuring the integrity of the Casparian strip formation in Arabidopsis roots. Previous evidence suggests that the SGN pathway may also be involved in root-microbe interactions, K sensing and more globally in the systemic integration of root responses. Given the essential role of the Casparian strip in the selective absorption of nutrients and water and the key role of the SGN pathway in ensuring its proper formation and integrity, it is difficult to disentangle the alternative roles of the SGN pathway from its function in CS formation. In this study, the authors aim to solve this by uncoupling the barrier formation function of the SGN pathway to explore additional roles of SGN signalling.

The trick used is to create a genetic feedforward loop by driving the expression of MYB36, the key transcription factor for CS formation, from the promoter of one of its targets (pCASP1). The resulting runaway behaviour leads to the premature formation of a wider CS independently able to bypass the requirement for SGN signalling to complete CS closure (MYB36Loop in *sgn3-3*). The authors then evaluate the MYB36Loop plants (not the MYB36Loop/*sgn3-3*) line with respect to a range of conditions mimicking plants' responses in more agricultural-relevant settings than most studies so far. They carefully monitor root architecture, abiotic stress response, microbial communities, and shoot development on different soils in response to nitrogen fertilisation. They conclude that independently from its CS surveillance function, SGN signalling integrates soil (a)biotic status between above- and belowground tissues.

The authors nicely combine cell biological analyses with physiological responses and microbiome analysis. The manuscript is original, well-written, illustrated, and enjoyable to read. Overall, this is a very good manuscript. Although I understand the motivation and rationale of the work: study the role of the SGN pathway in plant response independently from its role in ensuring the integrity of the CS, I need to be convinced that the work presented here is actually successfully tackling this. Here, the authors engineered plants that prematurely formed a wider CS and did so by bypassing the SGN pathway. Most analyses

compare these plants to myb36 plants that do not form a functional CS. As the motivation is to disentangle the function of the SGN pathway from its role in CS formation, why not compare the responses of the MYB36Loop to the one of the MYB36Loop/sgn3-3. As a geneticist, I think only this comparison would allow pinpointing the SGN-dependent responses independent of barrier formation. It could be that I am missing a key element, but then the authors should clarify this point.

Minor points:

- p2 l30: "surveillance" → "surveillance"
- p4 l26: "Indeed, plants expressing MYB36Loop had an almost tripled the anticlinal CS width" remove 'the'

Referee #2:

In this manuscript, the authors describe the generation of a specific and novel Casparian Strip mutant which allows to separate the Schengen signalling pathway from the CS formation process. As such, novel Schengen pathway functions can be uncovered. This reviewer really does like the approach and for sure there are very interesting novel insights in this story. The writing can however be improved somewhat as there are quite a lot of typo's and smaller grammar issues throughout the text.

Intriguingly, barrier formation in plants acts at the interface between development, interactions with the soil, nutrient transport and microbiome interactions. This makes the topic very interesting and relevant to study. I do feel however that in their holistic approach trying to connect all these things together, this manuscript fails to provide a clear storyline which would be easy understand by a non-expert reader. Although all experiments are by themselves interesting, properly executed and merit publication, the combination of all does result in a sort of patchwork of information which makes it challenging to follow the storyline and see why exactly these experiments are done. As one example, the microbiome study is interesting for sure and yields nice results. I fully understand why this is done and how it is connected to the rest, but all these different lines of evidence make it impossible to comprehend e.g. the final model the authors propose. I would really suggest to slim down the story to a more logical set of experiments which lead to a clear message at the end. For sure all these topics (shoot status, CS status, soil status, root response and root microbiota) will contribute to the final output, but currently there are so many parameters involved that the conclusions drawn from the data are complex and speculative. I understand this main comment is by itself also somewhat vague, but I am sure the authors will understand my point and likely had the same feeling already. I do believe there is plenty of interesting lines of evidence in the manuscript which would allow to slim down the story to a more comprehensive message at the end. If this means some parts are omitted, this reviewer is perfectly fine with it.

Some main and smaller comments are discussed in more detail below. Please note that the number of comments decreases as the story progresses, mainly as it became increasingly difficult to integrate all these different levels of regulation in my mind and follow the logic of the story. This should be improved.

- Line9: is the pCASP1-MYB36 (MYB36loop) construct in a WT or myb36 mutant background? If in a WT background, would a myb36 mutant background not be even better? Related, in Fig1A: what is "WT" in the WT MYB36loop sample? The authors should better explain in the text what these lines are as they form the basis of the story.
- Line26: does the increase in the length of the zone where the CS is deposited really indicate an increased CS deposition, or rather that the guiding machinery is affected? If the deposition machinery is affected, I would expect a thicker CS. When guidance is affected, I can expect an ectopic but correct deposition. Can the authors comment on this a perhaps make the description in the text more correct?
- Line 32 and Fig 1E: if you have the statistics done, there is no place for subjective statements like "slightly earlier". From your analysis, #1 is not significantly different from #5. Line #5 is significantly different from the WT and line #1 is not. As such, the two lines showing a similar transcriptional response in e.g. Fig 1A are not showing the same phenotype in Fig 1E. Likely the authors can perform more replicates of the experiment to also get line #1 to be significantly different. It would be good to have two independent lines showing the same (significant) effect. I understand this is a bit of a semantics discussion, but otherwise why bother doing statistics to begin with.
- Line 33: is it possible that the increased length of the CS deposition is more efficient in blocking PI penetration and as such, it is not faster maturation per se, but rather a more pronounced barrier with the same developmental stage as the WT? Can the authors comment on this alternative hypothesis?
- Line 15, Line 27 p6: please refrain from using subjective adjectives like "slightly" and describe the results in light of the statistical analysis which was performed.

- Line 12-14 p7: I do apologise for pointing this out, but according to the statistics on Fig 3F, there are no significant differences in Col-0 plants on MS, low N or no N as they all have overlapping significance groups. As such, the statement on these lines cannot be made. The only significant difference in this graph within a genotype is seen in WT MYB36loop plants for no N

compared to MS and low N. Given these phenotypes (and in other graphs) are borderline significant (but I agree there is a trend), perhaps the number of samples needs to be increased here. As a more general comment: have these experiments been repeated? I can only see graphs with e.g. $n=7$ and no indication of biological repeats in the text, figure or legends.

- Fig4A: there is no indication of the asterisk in the legend. Assuming this is a statistical test, why is this different from the others performed. On line 28 p7, it is mentioned there is no increase in the myb36 line, but this seems also increased (less than the #5 line); but without proper statistical analysis, we cannot draw conclusions here. Related, why is the variability in the data much smaller for WT and #5 compared to Col-0 and myb36?

- Line 7-8 p8: What is the evidence that the reduced complexity but higher load of the bacterial and fungal community would be a less 'healthy root microbiota'? From the data in previous figures, one would argue that the MYB36loop plants are e.g. performing better under stress conditions. How would this changed microbiota then be a bad thing? They are changed for sure, but how do the authors reach the conclusion that this is less healthy?

Smaller remarks:

- Fig1F: why is PI penetration expressed in % of root in panel F and as cells after elongation zone in E?

- Fig3C: some of the text on the x-axis is missing, 3E: N is not indicated.

- Line 30 p7: it would be useful to explain alpha and beta diversity to the non-expert reader.

Referee #1:

Comment: The authors nicely combine cell biological analyses with physiological responses and microbiome analysis. The manuscript is original, well-written, illustrated, and enjoyable to read. Overall, this is a very good manuscript. Although I understand the motivation and rationale of the work: study the role of the SGN pathway in plant response independently from its role in ensuring the integrity of the CS, I need to be convinced that the work presented here is actually successfully tackling this. Most analyses compare these plants to *myb36* plants that do not form a functional CS. As the motivation is to disentangle the function of the SGN pathway from its role in CS formation, why not compare the responses of the MYB36Loop to the one of the MYB36Loop/*sgn3-3*. As a geneticist, I think only this comparison would allow pinpointing the SGN-dependent responses independent of barrier formation. It could be that I am missing a key element, but then the authors should clarify this point.

Response: *Thank you for this comment, we are happy to hear that the reviewer enjoyed our efforts. Overall, we agree that genetically, it makes sense to make this comparison and we have now included these lines into our soil experiments (figure S3G and figure S4). However, below we explain our reasoning for not doing this in the first place.*

*Our first observation was in the root transcriptome analysis where SGN-related responses were reduced in the MYB36_{Loop} roots (Fig 1 and suppl. Fig 1, repression of CASP4 as the only gene that overlaps *myb36* and SGN3 responses), which implies that the MYB36Loop line contains a deactivated SGN system. However, this is not conclusive and, in our opinion, the best evidence for this would be to create a mechanism to activate the endogenous SGN pathway independent of the CS and show this would still be possible in the MYB36Loop lines but not in the *sgn3-3* background. Unfortunately, this is not technically feasible as we lack a dominantly active SGN3 receptor.*

*What also convinced us for this overlap in function, was that the transcriptional responses in the shoots of the *sgn3-3* mutant overlapped with those of the MYB36_{Loop} line despite their opposing status of CS (defective in *sgn3-3*, enhanced in the MYB36_{Loop} line). This strongly indicates that the observed effects are due to the SGN-signaling and not directly the CS. To substantiate this, we have now included qPCR analysis to show that these genes show similar response in the *myb36loop sgn3-3* background (supl fig 3).*

*We hope the reviewer agrees with us in this and would further also like to emphasize an additional point which support our hypothesis. In this preprint (doi.org/10.1101/2023.12.06.570432, Figure 2),, we were able to demonstrate that a non-native SGN3 (*Lotus japonicus* SGN3) cannot restore CS function in *Atsgn3-3* roots, but can restore nitrogen response in *Atsgn3-3* shoots. This nicely complements with the findings of *sgn3-3/MYB36_{Loop}* lines presented here, showing that SGN3 can integrate nitrogen status in the soil with shoot response independently of its function in CS establishment.*

Minor points:

- p2 l30: "surveillance" → "surveillance"

- p4 l26: "Indeed, plants expressing MYB36Loop had an almost tripled the anticlinal CS width" remove 'the'

Response: *Thank you for pointing this out, we have corrected these mistakes.*

Referee #2:

Comment: In this manuscript, the authors describe the generation of a specific and novel Casparian Strip mutant which allows to separate the Schengen signaling pathway from the CS formation process. As such, novel Schengen pathway functions can be uncovered. This reviewer really does like the approach and for sure there are very interesting novel insights in this story. The writing can however be improved somewhat as there are quite a lot of typo's and smaller grammar issues throughout the text.

Response: *We thank the reviewer for the excitement about our work and the keen eye on the language. We apologize for the grammar issues. It is always difficult as a non-native English speaker to catch all grammatical errors. We have gone through the manuscript several times, with specific focus on this and hope to have weeded out these mistakes.*

Comment: I do feel however that in their holistic approach trying to connect all these things together, this manuscript fails to provide a clear storyline which would be easy understand by a non-expert reader. Although all experiments are by themselves interesting, properly executed and merit publication, the combination of all does result in a sort of patchwork of information which makes it challenging to follow the storyline and see why exactly these experiments are done. As one example, the microbiome study is interesting for sure and yields nice results. I fully understand why this is done and how it is connected to the rest, but all these different lines of evidence make it impossible to comprehend e.g. the final model the authors propose. I would really suggest to slim down the story to a more logical set of experiments which lead to a clear message at the end. For sure all these topics (shoot status, CS status, soil status, root response and root microbiota) will contribute to the final output, but currently there are so many parameters involved that the conclusions drawn from the data are complex and speculative.

Response: *Thank you for this constructive comment. We agree that the manuscript indeed felt somewhat incoherent. We think the reason for this was our attempt to put this new CS model into a complete context of the current set of barrier mutants. However, since there is a lot of work done on this subject, we agree that our attempt has rather hindered the story than made it useful. We put a lot of thoughts into how to improve this and in our opinion, the microbiome analysis (Fig4) give rise to most confusion in the overall model. We decided to remove this part from the manuscript. Our reasoning for this was that despite the very interesting findings (as the reviewer also points out) we think this is the main cause for distraction from the physiological*

functions typically associated with root barriers. By removing this part, we have simplified our focus to be on the mechanistic role of the CS in providing quantitative SGN output (see Figure 6). We aim to shape the microbiome findings up individually in combination with our recent findings in *Lotus japonicus* (see comment to reviewer 1) and hope that the reviewer agrees to this decision.

Comment: Some main and smaller comments are discussed in more detail below. Please note that the number of comments decreases as the story progresses, mainly as it became increasingly difficult to integrate all these different levels of regulation in my mind and follow the logic of the story. This should be improved.

Response: *We hope that the updated manuscript and also the model in figure 6 has made this clearer.*

Comment: - Line9: is the pCASP1-MYB36 (MYB36loop) construct in a WT or myb36 mutant background? If in a WT background, would a myb36 mutant background not be even better? Related, in Fig1A: what is "WT" in the WT MYB36loop sample? The authors should better explain in the text what these lines are as they form the basis of the story.

Response: *In a myb36 KO background, the MYB36_{Loop} will not be activated since the activation depends on the endogenous role of MYB36 to establish the CSD in the first place via activation of i.e. the CASP1 promoter. We agree with the reviewer that it is important to use proper lines for comparison, we therefore used Col-0 for myb36-2 and all of our MYB36_{Loop} lines are made in a pCASP1:CASP1-GFP background from Roppolo et al 2011 as these are the respective parental lines. We consistently refer to the pCASP1::CASP1-GFP background as "WT" . We have added information on genetic backgrounds in the figures and legends for clarification. Please also see comments for reviewer 1 regarding genetic backgrounds.*

Comment: - Line26: does the increase in the length of the zone where the CS is deposited really indicate an increased CS deposition, or rather that the guiding machinery is affected? If the deposition machinery is affected, I would expect a thicker CS. When guidance is affected, I can expect an ectopic but correct deposition. Can the authors comment on this a perhaps make the description in the text more correct?

Response: *We think it's likely to tie into both. The nature of the increased transcriptional responses observed in the MYB36loop plants would imply that both the timing as well as "dose" of CS forming machinery is increased exponentially, at least initially upon activation of the loop. Under "normal" developmental conditions, these responses would be switched off again by unknown mechanism(s) to coordinate the correct CS formation and timing. However, the "run-away" nature of the loop decouples these and thereby overloads the machinery that is responsible for deposition and guiding the CS. We believe this can actually serve as an intriguing cell biological tool to tease out which mechanisms may be involved subcellularly. We have implemented these thoughts into the discussion P.10 L. 10-17.*

Comment: - Line 32 and Fig 1E: if you have the statistics done, there is no place for subjective statements like "slightly earlier". From your analysis, #1 is not significantly different from #5. Line #5 is significantly different from the WT and line #1 is not. As such, the two lines showing a similar transcriptional response in e.g. Fig 1A are not

showing the same phenotype in Fig 1E. Likely the authors can perform more replicates of the experiment to also get line #1 to be significantly different. It would be good to have two independent lines showing the same (significant) effect. I understand this is a bit of a semantics discussion, but otherwise why bother doing statistics to begin with.

Response: *The reviewer is right, there is no room for subjective degrees of statistical significance under set thresholds. We have repeated this experiment and measured ratios instead of cell number similar to Fig 1E. It was indeed confusing that these measurements were done in different ways, despite the conclusions being similar. We have repeated and normalized this in all relevant experiments throughout the manuscript as well as rephased all statements.*

Comment: - Line 33: is it possible that the increased length of the CS deposition is more efficient in blocking PI penetration and as such, it is not faster maturation per se, but rather a more pronounced barrier with the same developmental stage as the WT? Can the authors comment on this alternative hypothesis?

Response: *This comment relates to the one above regarding deposition of CS (line 26). The reviewer is correct that this could be an alternative hypothesis and we have included this in the discussion (P.10 L10-17). According to Naseer et al 2012, PNAS there is a delay in PI blockage when compared to CS autofluorescence. This is likely due to the difference between CS initiation and functional establishment in WT plants.*

Comment: - Line 15, Line 27 p6: please refrain from using subjective adjectives like "slightly" and describe the results in light of the statistical analysis which was performed.

Response: *Please see our response to the similar comment above. We have repeated all these experiments and additionally added MYB36_{Loop} lines in the sgn3-3 background according to the request from reviewer 1.*

Comment: - Line 12-14 p7: I do apologise for pointing this out, but according to the statistics on Fig 3F, there are no significant differences in Col-0 plants on MS, low N or no N as they all have overlapping significance groups. As such, the statement on these lines cannot be made. The only significant difference in this graph within a genotype is seen in WT MYB36loop plants for no N compared to MS and low N. Given these phenotypes (and in other graphs) are borderline significant (but I agree there is a trend), perhaps the number of samples needs to be increased here. As a more general comment: have these experiments been repeated? I can only see graphs with e.g. n=7 and no indication of biological repeats in the text, figure or legends.

Response: *No need for the reviewer to apologize, we appreciate the detailed attention to the statistical analysis. Our point here was to emphasize the differences in the MYB36_{Loop} lines and its corresponding WT parental line. To make this clear, we have performed an additional experiment, compiled the two, emphasized the individual datapoints and changed the measurement to % of WT blockage to highlight the differences of interest. In fact, since the conclusion of the differences between the individual lines was not our main point and has been addressed elsewhere, we argue that a Students T-test between full treatment (MS) and the reduced N treatments is sufficient for our conclusion. Moreover, in this compiled experiment, we do indeed see a significant difference already at low (0.1 mM) N in the media in both lines.*

Comment: Fig4A: there is no indication of the asterisk in the legend. Assuming this is a statistical test, why is this different from the others performed. On line 28 p7, it is mentioned there is no increase in the myb36 line, but this seems also increased (less than the #5 line); but without proper statistical analysis, we cannot draw conclusions here. Related, why is the variability in the data much smaller for WT and #5 compared to Col-0 and myb36?

Response: *We thank for this comment, but since the fig 4 has been taken out, we will keep this in mind if we use this data elsewhere.*

Comment: - Line 7-8 p8: What is the evidence that the reduced complexity but higher load of the bacterial and fungal community would be a less 'healthy root microbiota'? From the data in previous figures, one would argue that the MYB36loop plants are e.g. performing better under stress conditions. How would this changed microbiota then be a bad thing? They are changed for sure, but how do the authors reach the conclusion that this is less healthy?

Response: *We thank for this comment, but since the fig 4 has been taken out, we will keep this in mind if we use this data elsewhere.*

Smaller remarks:

- Fig1F: why is PI penetration expressed in % of root in panel F and as cells after elongation zone in E?

Response: *We have solved this issue by repeating the experiment. See comment above.*

- Fig3C: some of the text on the x-axis is missing, 3E: N is not indicated.

Response: *We apologize for this and have included this information in the current version.*

- Line 30 p7: it would be useful to explain alpha and beta diversity to the non-expert reader.

Response: *We thank for this comment, but since the fig 4 has been taken out, we will keep this in mind in case we use this data elsewhere. We hope the reviewer agrees with this decision despite a number of interesting findings came from the microbiome analysis.*

Dear Tonni,

Thank you submitting a revised version of your manuscript. It was sent to the same reviewers that originally appraised your work; their comments are attached to the bottom of this email. As you will see, both are satisfied with the changes you made. Before we can move forwards towards publication of your manuscript, though, there are some remaining editorial points which need to be addressed. In this regard, would you please:

- select five keywords,
- change the title of the 'Conflict of Interests' statement to the 'Disclosure and Competing Interests Statement',
- remove the author credit section from the manuscript,
- upload figures as individual files without legends; these should only be included in the main manuscript file,
- rename Tables EV1-EV3 as Dataset EV1-EV3 with the corresponding callouts in the text. Remove legends for these figures from the main manuscript file and included them a separate tab in each Excel file; Table EV4 should be renamed as Table EV1 with the appropriate callout and legend included in the Excel file,
- provide source data for Fig. 4E; source data files need to be saved in a scheme one figure/folder and then uploaded as .zip files. All the Source data files for figure 1, for example, need to be saved in a single folder and this needs to be zipped and then uploaded as a "SD figure 1.zip" file,
- a plant image is re-used between examples shown in Figure 5A and Fig EV4; check these images for inadvertent duplication,
- ensure that dataset PRJNA940103 is fully publically available upon final acceptance of this manuscript and provide a URL in the data availability section of the manuscript,
- consider including a separate 'Data Information' section in the legends of figures 1a, c, e-f; 3a-c, e-f; 4b, e; 5b-c; EV 1d, h; EV 2d; EV 3b, g,
- rectify figure 4b, which is currently incorrectly labelled as 4a,
- there are no box plots in the figure EV 1; rectify the box plot related information in the figure legend appropriately,
- define n in the legend of figure EV 3f,
- define error bars in the legend of figure EV 1h,
- define the measure of center for the error bars in the legends of figure EV 3f,
- include a scale bar in figure EV 4,
- define the scale in figure 2a and EV 1f-g,
- rename movie as Movie EV1 with the corresponding callout, remove the legend from the ms file, and zip with the movie file, and
- place main and EV figure legends below the References

We include a synopsis of the paper (see <http://emboj.embopress.org/>). Please provide me with a two-sentence general summary statement and 3-5 bullet points that capture the key findings of the paper.

We also need a summary figure for the synopsis. The size should be 550 wide by [200-400] high (pixels). You can also use something from the figures if that is easier.

I look forward to receiving these changes. EMBO Press is an editorially independent publishing platform for the development of EMBO scientific publications.

Best wishes,

William

William Teale, PhD
Editor
The EMBO Journal
w.teale@embojournal.org

We realize that it is difficult to revise to a specific deadline. In the interest of protecting the conceptual advance provided by the work, we recommend a revision within 3 months (12th Jun 2024). Please discuss the revision progress ahead of this time with the editor if you require more time to complete the revisions. Use the link below to submit your revision:

Referee #1:

I thank the authors for their detailed explanation of the rationale of their study. I am now convinced. I agree that removing the microbiome part streamlines the message.

Referee #2:

After reading through the new version of the manuscript and the reply to reviewer comments, I am satisfied with the changes made to the manuscript. I would also like to applaud the authors for the way they have taken up the suggestions and comments in a very positive and constructive manner. I can only hope the authors will agree with me that the reviewing process has made this into a better manuscript.

- Please reduce the number of keywords on the abstract page to five (ideally choosing broad general terms).

We have changed this accordingly.

- The legends are missing in Dataset EV2-EV3 - please include the legends in the corresponding Datasets as a separate tab in each Excel file

We have created a new sheet named as Legend in Dataset EV2 and EV3.

- Please remove the "Supplementary information" section (including Dataset legends) from the manuscript file

We have changed this accordingly.

- Please zip the movie legend (in .docx format) with the movie file and upload it as a zipped file "Movie EV1"

We have changed this accordingly.

- Please remove the Synopsis text from the manuscript file, and upload it as a separate file (in .docx format as Cover Art/Synopsis Image)

We have changed this accordingly.

- The size of the Synopsis image should be (exactly) 550 pixels wide and 300-600 pixels high. Please resize the provided image which is currently 695x528.

We now provide a JPG image with 550 pixels width and 418 pixels height.

Dear Tonni,

I am pleased to inform you that your manuscript has been accepted for publication in the EMBO Journal.

We made it! Many thanks for choosing The EMBO Journal - I will be delighted to see this in our pages.

Yours sincerely,

William

William Teale, PhD
Editor
The EMBO Journal
w.teale@embojournal.org
